# Matching High-Dimensional Geometric Quantiles for Test-Time Adaptation of Transformers and Convolutional Networks Alike

## Abstract

Test-time adaptation (TTA) refers to adapting a classifier for the test data when the probability distribution of the test data slightly differs from that of the training data of the model. To the best of our knowledge, most of the existing TTA approaches modify the weights of the classifier relying heavily on the architecture. It is unclear as to how these approaches are extendable to generic architectures. In this article, we propose an architecture-agnostic approach to TTA by adding an adapter network pre-processing the input images suitable to the classifier. This adapter is trained using the proposed *quantile loss*. Unlike existing approaches, we correct for the distribution shift by matching high-dimensional geometric quantiles. We prove theoretically that under suitable conditions minimizing quantile loss can learn the optimal adapter. We validate our approach on CIFAR10-C, CIFAR100-C and TinyImageNet-C by training both classic convolutional and transformer networks on CIFAR10, CIFAR100 and TinyImageNet datasets.

## 1 Introduction

In the past decade, deep learning approaches have achieved great progress in solving the image classification problem. Most of these approaches operate under the assumption that the probability distribution of the images at test-time (a.k.a. target) is identical to that of the training (a.k.a source). While these networks perform well on the source data, they achieve significantly lower accuracy on the target data if the target data consists of images with commonly caused weather or sensor degradations. As such corruptions occur in real-world applications, it is necessary to build image classifiers that can adapt to the test-time distribution.

Datasets such as MNIST-C, CIFAR10-C, CIFAR100-C, TinyImageNet-C, ImageNet-C Hendrycks & Dietterich (2019), etc simulate the commonly occurring degradations at various levels of severity. These datasets have served as a test-bed for the checking if the classifiers can adapt to covariate shifts. Test-time adaptation (TTA) refers to building classifiers that adapt to the test data assuming access to 1) the classifier trained on the source data, and 2) the target data. The simplest setting - the target distribution is static and is different from the training distribution. In the literature, several works have addressed other settings of TTA such as 1) the type and severity of the corruption in the target distribution dynamically changes with time Wang et al. (2022); Brahma & Rai (2023), 2) additionally the test-time distribution is $\epsilon$-contaminated Gong et al. (2024) etc.

Existing approaches obtain good results on the benchmark datasets. Some notable approaches include BNStats Nado et al. (2020), TENT Wang et al. (2020), CoTTA Wang et al. (2022), EATA Niu et al. (2022), SoTTA Gong et al. (2024), RoTTA Yuan et al. (2023), SAR Niu et al. (2023), MedBN Park et al. (2024) etc. However, they seem to be applicable only to CNN-based classifiers such as ResNet-like architectures. BNStats, MedBN, RoTTA, SoTTA and TENT explicitly assume that the underlying pre-trained classifier contains batch-norm layers. The authors of SAR report that their approach suffers when group and layer norms are present in the architecture. Group-norm (a generalization of layer-norm and instance-norm) Wu & He (2018) stabilizes internal representations for training with small batches. Covariate shift is a global

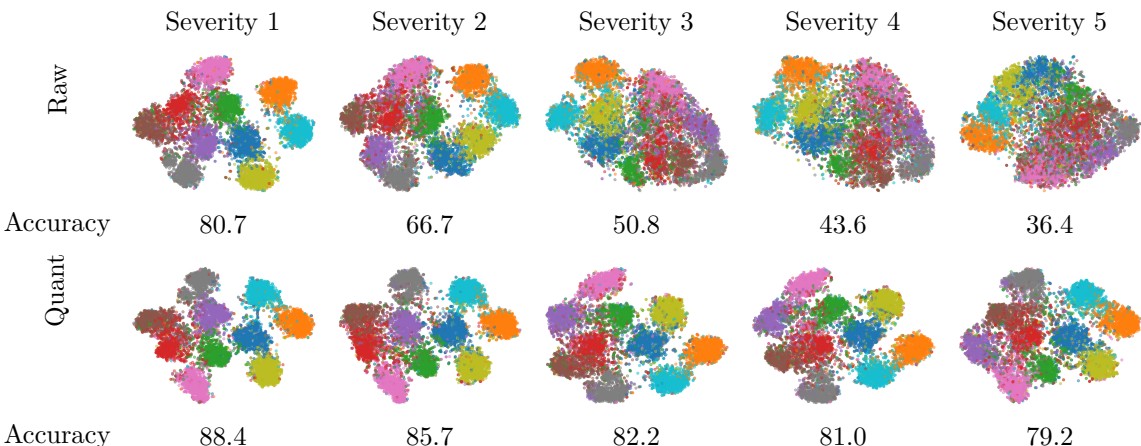

Figure 1: Comparing t-SNE plots of raw ResNet18 features versus Quantile-Loss corrected ResNet18 features on CIFAR10C with Gaussian distortions. ResNet18 is trained on CIFAR10. Observe that the corrected features maintain the clusters well at higher levels of distortions while the raw features lose the class structure as severity increases.

semantic drift that normalization alone cannot fix as they alter statistics beyond means and variances. EATA and CoTTA do not require batch-norm layers. However, EATA obtains much lower performance compared to the others. On the other hand, CoTTA uses a mean-teacher approach i.e. requires two copies of the classifier and is memory-inefficient. Retraining of classifiers is both memory and computationally inefficient when activations take up a lot of GPU memory and the training requires large volumes of data. Hence, they are not easily extendable to newer architectures such as vision transformers (ViTs) which are the current state-of-the-art. We thus pose and answer the question - *Is it possible to design approaches that are architecture-agnostic?*.

Our solution is to set up an architecture-agnostic design by avoiding the modification of the classifier. To validate our solution, we work with the simplest setting i.e. the target distribution is different and static. We make an additional assumption - access to the features generated by the classifier of the unlabeled source data. This is a reasonable assumption for many applications. We do not require the annotations of the train or the target data. Since the target data distribution is assumed to be a corruption of the source data, we assume the existence of a mapping (de-corruption operator) that takes a corrupted image as an input and outputs an image from the distribution of the source data. Fig 1 shows the comparison of the t-SNE Maaten & Hinton (2008) plots of the CIFAR10C features obtained by ResNet18 (trained on CIFAR10) against the t-SNE plots of decorrupted counterparts. The decorruptor is trained by freezing the classifier and minimizing *quantile loss* in the feature space. Observe that the cluster structures are well-preserved when decorrupted. On the other hand, as the severity increases, the uncorrected raw features lose the cluster structure.

Our contributions are summarized as follows:

1. We propose an architecture-agnostic approach to solve TTA by explicitly learning a de-corruption mapping between the target and the source image spaces. The classifier pre-trained on the sourced data is frozen. The solution is to minimize a novel loss function *quantile loss* between the features of the source data and the features obtained from the decorrupted target images (see Fig 2). The *quantile loss* is based on high-dimensional geometric quantiles and quantifies closeness between high-dimensional probability distributions. At inference, the decorrupter model is applied to the test image and the classifier is applied to the output. This eliminates modification or fine-tuning of large classifier models such as ViTs.

2. We show the existence of a 'good' initialization of the decorruptor architecture at which minimization of quantile loss is theoretically equivalent to training the decorruptor using corrupt-clean pairs of

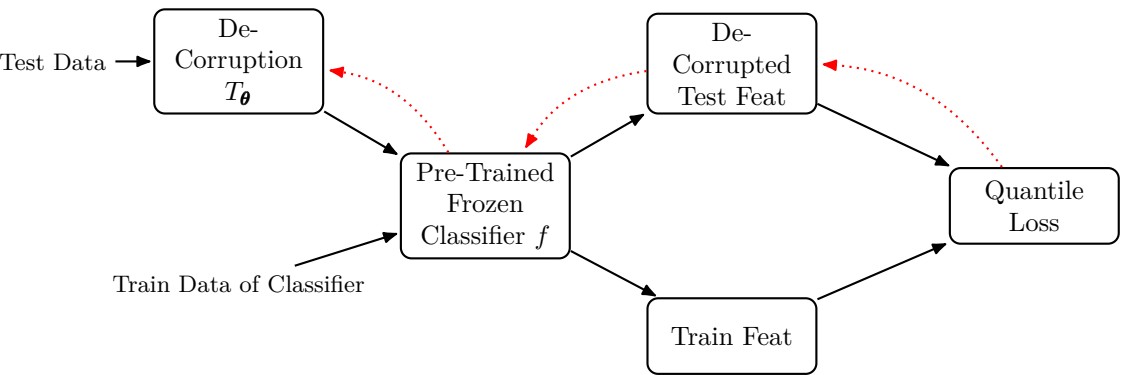

Figure 2: Quantile-Matching pipeline - The black arrows denote the forward propagation. The dotted red arrows denote the backpropagation.

     images - which is a proxy for aligning class-conditionals. Thus, under 'good' initialization, matching marginals of the features by minimizing quantile loss also aligns class-conditional features eliminating the need for any label information. Thanks to the non-trivial performance of classifiers on covariate-shifted data, an identity map satisfies the 'good' initialization property.

3.  The *quantile loss* is not sample-separable but admits a weak-separability property dubbed as composite-separability. We propose an efficient algorithm for small-batch training of composite-separable loss functions using a memory bank.

4.  We validate our claim on architecture-agnostic approach by showing empirical evidence on standard benchmark datasets CIFAR10C, CIFAR100C and TinyImageNetC using ResNet18, a compact convolutional transformer (CCT), convolutional vision transformer (CVT) and a light weight vision transformer (ViT-Lite).

The rest of the article is organized as follows-: in section 2, we formally state the TTA problem and summarize the existing approaches. We briefly discuss as to why the current approaches either cannot be adapted to newer architectures or are inefficient. In section 3, we provide a background of high-dimensional geometric quantiles including a few existing results. We then define the quantile loss along with geometric intuition behind its design along with practical considerations on its usage. We then describe quantile-loss based solution to solving TTA and describe our algorithm. We characterize the class of loss functions that admit the weak sample-separability property that the quantile loss satisfies and propose an efficient algorithm to implement SGD for small batch-sizes. In section 5, we validate our claims by applying quantile loss to both convolution-based classifiers and transformer-based classifiers for improving accuracy of the baselines. We then provide ablation studies on the choice of hyperparameters and provide pointers on choosing them.

## 2 Problem Setup and Existing Approaches

### 2.1 Problem Setup

The benchmarking datasets typically assume only a special case of TTA namely covariate-shifts. We thus formally define the TTA specific to covariate-shifts:

**Notation**: Let $\mathcal{D} = \{(\boldsymbol{x}_i, y_i)\}_{1 \leq i \leq n}$, $\mathcal{D}^+ = \{(\boldsymbol{x}_j^+, y_j)\}_{1 \leq j \leq m}$ be sample data from the clean joint probability distribution $\mu_{\boldsymbol{X},Y}$ and corrupted joint distribution $\nu_{\boldsymbol{X},Y}$ respectively. Covariate-shift allows us to assume a joint distribution on $(\boldsymbol{X}^+, \boldsymbol{X})$ such that $\boldsymbol{X} = T^*(\boldsymbol{X}^+) + \boldsymbol{\epsilon}$ with $E[\boldsymbol{\epsilon}|\boldsymbol{X}] = \boldsymbol{0}$ and $\|Cov(\boldsymbol{\epsilon}|\boldsymbol{X})\|_{op} \leq \delta$ i.e. the highest Eigenvalue of the covariance matrix is bounded by $\delta$. Here $y \in \mathbb{R}$ denotes the class-labels of the features $\boldsymbol{x}, \boldsymbol{x}^+ \in \mathbb{R}^d$. Let $T_{\boldsymbol{\theta}} : \mathbb{R}^d \to \mathbb{R}^d$ with $\boldsymbol{\theta} \in \Theta$ denote a family of functions. Let $\nu_{\boldsymbol{\theta},\boldsymbol{X},Y}$ denote the de-corrupted joint distribution at parameter $\boldsymbol{\theta}$ i.e. distribution of $(T_{\boldsymbol{\theta}}(\boldsymbol{X}^+), Y)$ with $(\boldsymbol{X}^+, Y) \sim \nu_{\boldsymbol{X},Y}$. Let $f : \mathbb{R}^d \to \mathbb{R}^k$ denote the feature extractor of the classifier trained on $\mu_{\boldsymbol{X},Y}$. Let the distribution of features

obtained from the clean and the de-corrupted distributions at parameter $\boldsymbol{\theta}$ using the frozen classifier be denoted by $\mu^f = f_\# \mu_{\boldsymbol{X}}$ and $\nu_{\boldsymbol{\theta}}^f = f_\# \nu_{\boldsymbol{X},\boldsymbol{\theta}}$ respectively. Let $h : \mathbb{R}^k \to \mathbb{R}^c$ denote the fully connected layer (i.e. last linear layer) of the classifier. Here $c$ denotes the number of classes.

**Problem Statement**: Can we correct for the covariate shifts i.e. estimate the de-corruption operator $T^*$ so that the classifier $f$ can be applied on the de-corrupted data? Note that we have access only to the $k$-dimensional features generated by the unlabelled original data - $\{f(\boldsymbol{x}_i)\}_{1 \le i \le n}$, and the unlabelled corrupted images - $\{\boldsymbol{x}_j^+\}_{1 \le j \le m}$, ?

**Remarks:** The requirement on access to the features generated by the classifier on the training data is a mild assumption. Privacy concerns on the training data does not arise as our method requires only the probability distribution of these features. While the existing approaches do not use these features in their methods, we claim that this information is useful at the cost of a small memory overhead for storing the source features.

### 2.2 Literature Review

Existing approaches for TTA typically require either 1) modification of the batch-norm layers, or 2) retraining/fine-tuning of the classifier. Broadly, they use five categories of ideas: 1) batch-norm modification, 2) mean-teacher retraining, 3) Fisher-information regularization, 4) sharpness-aware entropy minimization, and 5) pseudo-label generation

**BN-dependent methods** These methods assume that the classifier uses batch-norm layers and adapt the running statistics at the test time. BNStats Nado et al. (2020) replaces the batch-norm statistics with those estimated from the test data. MedBN Park et al. (2024) uses the median of the test statistics instead of the mean. RoTTA Yuan et al. (2023) employs an exponential moving average (EMA) of BN statistics from test batches, along with a teacher-student retraining scheme and a memory bank that favors recent and confident samples. SoTTA Gong et al. (2024) is similar to RoTTA but additionally optimizes a combination of entropy and perturbation entropy to avoid sharp minima. These approaches are effective when batch-norm layers are present but do not directly extend to transformer architectures, which typically use layer-norm and group-norm layers instead of batch-norm.

**Mean-teacher retraining methods** These methods duplicate the model and use consistency training between a teacher and student. RoTTA uses slow EMA updates for the teacher and fast updates for the student. CoTTA Wang et al. (2022) generates multiple augmentations and uses a majority-voting of the predicted labels whenever the model is under-confident. Additionally, it resets a fraction of parameters to the original state to reduce catastrophic forgetting. Since these methods are not inherently tied with batch-norm, they could be adapted to transformers in principle, but computational cost and training stability remain as challenges.

**BN-independent regularization methods** These methods adapt models through entropy-based objectives and information-theoretic constraints. TENT Wang et al. (2020) minimizes prediction entropy during test time. SAR Niu et al. (2023) minimizes a combination of entropy and perturbation entropy. It is however reported to degrade in the presence of group-norm and layer-norm layers, making transformer adaption non-trivial. EATA Niu et al. (2022) minimizes entropy with a Fisher-information regularizer. It is sample-efficient and scalable to long test streams, but on CIFAR100C its performance lags behind all other approaches.

## 3 Quantile Loss: A Proxy for Matching High-Dimensional Geometric Quantiles

### 3.1 Background Theory

Univariate quantiles can be viewed as solutions to a loss-minimization problem. Geometric quantiles are natural extensions of the loss-minimization problem in higher dimensions. They were introduced in Chaudhuri (1996). We recall the definition and replicate some of the useful properties here:

**Definition 1.** *Let $\boldsymbol{Z}_1, \cdots, \boldsymbol{Z}_n \in \mathbb{R}^k$ be k-dimensional vectors sampled from a common probability distribution. Let $B^{(k)} = \{\boldsymbol{u} : \boldsymbol{u} \in \mathbb{R}^k, \|u\|_2 < 1\}$ denote the Euclidean open ball of unit radius centered at the origin. For every $u \in B^{(k)}$ and $\boldsymbol{Q} \in \mathbb{R}^k$, define the loss function*

$$L_u(\boldsymbol{Z}_1, \cdots, \boldsymbol{Z}_n; \boldsymbol{Q}) = \frac{1}{n} \sum_{i=1}^{n} \Phi(\boldsymbol{u}, \boldsymbol{Z}_i - \boldsymbol{Q}) \tag{1}$$

*where $\Phi : \mathbb{R}^k \times \mathbb{R}^k \to \mathbb{R}$ is given by*

$$\Phi(\boldsymbol{u}, \boldsymbol{t}) = \|\boldsymbol{t}\|_2 + <\boldsymbol{u}, \boldsymbol{t}> \tag{2}$$

*with $\|.\|_2$ denoting the Euclidean norm and $< .,. >$ denoting the usual dot product. The geometric sample quantile at $\boldsymbol{u}$ is denoted by $\hat{\boldsymbol{Q}}_n(\boldsymbol{u})$ and is given by*

$$\hat{\boldsymbol{Q}}_n(\boldsymbol{u}) = argmin_{\boldsymbol{Q} \in \mathbb{R}^k} L_u(\boldsymbol{Z}_1, \cdots, \boldsymbol{Z}_n; \boldsymbol{Q}) \tag{3}$$

*Likewise, the geometric population quantile at $\boldsymbol{u}$ is defined as*

$$\boldsymbol{Q}(\boldsymbol{u}) = argmin_{\boldsymbol{Q} \in \mathbb{R}^k} E_{\boldsymbol{Z}} \left[ \Phi(\boldsymbol{u}, \boldsymbol{Z} - \boldsymbol{Q}) \right] \tag{4}$$

Some properties of geometric quantiles:

1. For every $\boldsymbol{u} \in B^{(k)}$, the sample quantile $\hat{\boldsymbol{Q}}_n(\boldsymbol{u})$ exists and is finite (as a consequence of the convexity of the loss function it minimizes).

2. For every $\boldsymbol{u} \in B^{(k)}$, the sample quantile $\hat{\boldsymbol{Q}}_n(\boldsymbol{u})$ is unique if $\boldsymbol{Z}_1, \cdots, \boldsymbol{Z}_n \in \mathbb{R}^k$ do not lie on a straight line. This assumption is reasonable for real data.

3. Every data sample $\boldsymbol{Z}_1, \cdots, \boldsymbol{Z}_n \in \mathbb{R}^k$ lies on the support of the sample quantiles of the distribution i.e. each $\boldsymbol{Z}_i$ is a sample quantile and the corresponding index $u_i \in B^{(k)}$ can be computed using theorem 1

**Theorem 1.** *(Inverse Map) Chaudhuri (1996): Consider the data $\boldsymbol{Z}_1, \cdots, \boldsymbol{Z}_n \in \mathbb{R}^k$ sampled from a common probability distribution. Assume that n is large and the samples do not lie on a straight line and are distinct. Let $\boldsymbol{u} \in B^{(k)}$. Suppose $\hat{\boldsymbol{Q}}_n(\boldsymbol{u})$ denotes the sample quantile (existence is guaranteed) then $\boldsymbol{u}$ satisfies the following relation:*

*If $\hat{\boldsymbol{Q}}_n(\boldsymbol{u}) \notin \{\boldsymbol{Z}_1, \cdots, \boldsymbol{Z}_n\}$ then*

$$\boldsymbol{u} = \frac{1}{n} \sum_{i=1}^{n} \frac{\hat{\boldsymbol{Q}}_n(\boldsymbol{u}) - \boldsymbol{Z}_i}{\|\hat{\boldsymbol{Q}}_n(\boldsymbol{u}) - \boldsymbol{Z}_i\|} \tag{5}$$

*and if $\hat{\boldsymbol{Q}}_n(\boldsymbol{u}) = \boldsymbol{Z}_r$ for some $1 \leq r \leq n$ then*

$$\boldsymbol{u} = \frac{1}{n-1} \sum_{i=1, i \neq r}^{n} \frac{\hat{\boldsymbol{Q}}_n(\boldsymbol{u}) - \boldsymbol{Z}_i}{\|\hat{\boldsymbol{Q}}_n(\boldsymbol{u}) - \boldsymbol{Z}_i\|} \tag{6}$$

Theorem 1 implies that every vector in $\mathbb{R}^k$ is a quantile of a probability distribution in $k$ dimensions (assuming the support is not on a straight line). The inverse map provides a canonical mapping between the space $\mathbb{R}^k$ and the open unit ball $B^{(k)}$.

**Theorem 2.** *(Population Quantiles Determine Probability Distribution Uniquely) Chaudhuri (1996) Let $\boldsymbol{U}, \boldsymbol{V}$ be random variables in $\mathbb{R}^k$ then if for each $\boldsymbol{u} \in B^{(k)}$ if the population quantiles are equal i.e. $Q_U(\boldsymbol{u}) = Q_V(\boldsymbol{u})$ then $\boldsymbol{U}$ and $\boldsymbol{V}$ have the same probability distribution.*

**Theorem 3.** *(Asymptotic Convergence) Chaudhuri (1996): Consider the data $\boldsymbol{Z}_1, \cdots, \boldsymbol{Z}_n \in \mathbb{R}^k$ sampled from a common probability distribution. Assume that n is large and the samples do not lie on a straight line and are distinct. Let $\boldsymbol{u} \in B^{(k)}$. Then the sample quantiles $\hat{\boldsymbol{Q}}_n(\boldsymbol{u})$ at $\boldsymbol{u}$ converge to the population quantiles $\boldsymbol{Q}(\boldsymbol{u})$ i.e.*

$$\hat{\boldsymbol{Q}}_n(\boldsymbol{u}) \to \boldsymbol{Q}(\boldsymbol{u}) \text{ as } n \to \infty \tag{7}$$

Theorem 2 and theorem 3 together imply that it if a pair of high-dimensional distributions $\boldsymbol{V}$ and $\boldsymbol{W}$ of the same dimension $k$ and we have i.i.d. samples $\{\boldsymbol{V}_i\}_{i=1}^n$ and $\{\boldsymbol{W}_j\}_{j=1}^m$ with $n$ and $m$ 'large' enough to approximate the population quantiles, it is enough to check if the sample quantiles of both the distributions are 'close'.

## 3.2 Quantile Loss: Measuring the Discrepancy Between High-Dimensional Distributions

**Core Idea**    The intuition is - if the estimate $T_{\boldsymbol{\theta}}$ of the de-corruption operator $T^*$ is accurate then the geometric high-dimensional quantiles of the features obtained from the classifier on the estimated de-corrupted images $f(T_{\boldsymbol{\theta}}(\boldsymbol{X}^+))$ and the features of the classifier on the original images $f(\boldsymbol{X})$ should be 'close' to each other.

The denoising operator $T_{\boldsymbol{\theta}}$ should be tuned such that the geometric quantiles of the two distributions are similar. Define the quantile index function $U_F : \mathbb{R}^k \to \mathbb{R}^k$ as

$$U_F(\boldsymbol{z}) = E_{\boldsymbol{X} \sim F}\left[\frac{\boldsymbol{z} - \boldsymbol{X}}{\|\boldsymbol{z} - \boldsymbol{X}\|}\right] \tag{8}$$

The quantile index function is well-defined and bounded in a unit ball $B^{(k)}$ in $\mathbb{R}^k$ as a consequence of Theorem 1. The quantile loss is defined as:

$$\mathcal{L}_{quant}(\boldsymbol{\theta}) = E_{\boldsymbol{Z} \sim \mu^f}\left[\|U_{\nu_{\boldsymbol{\theta}}^f}(\boldsymbol{Z}) - U_{\mu^f}(\boldsymbol{Z})\|^2\right] \tag{9}$$

The pseudo-code for quantile matching is given in Algorithm 1. The algorithm uses a full gradient descent to present the key ideas in a minimalistic way. In practice, the quantile indices are matched only on a subset of the support of the feature space of original images (see subsection 4.1 for details) to boost the results. Additionally, we work with mini-batches and maintain a memory bank for the estimated quantile indices for scaling up the algorithm with GPU constraints (subsection 4.2 contains more details).

---

**Algorithm 1:** Quantile Matching

**Input:** $\{\boldsymbol{x}_j^+\}_{1 \le j \le m}$: test data, $\{f(\boldsymbol{x}_i)\}_{1 \le i \le n}$: train features , $\boldsymbol{\theta}_0$: initial parameters to set an identity mapping $T_{\boldsymbol{\theta}_0}$, $N$: number of epochs

**Output:** $\boldsymbol{\theta}$: final optimized parameters

1 Pre-compute the quantile indices $\{\boldsymbol{u}_r\}_{r \in S}$ of quantiles $\{f(\boldsymbol{x}_r)\}_{1 \le r \le n}$ w.r.t. data $\{f(\boldsymbol{x}_i)\}_{1 \le i \le n}$;

2 Initialize $\boldsymbol{\theta} \leftarrow \boldsymbol{\theta}_0$ ;

3 **for** $l \leftarrow 1$ **to** $N$ **do**

4      Compute the estimated de-corrupted features at current parameters $\{f(T_{\boldsymbol{\theta}}(\boldsymbol{x}_j^+))\}_{1 \le j \le m}$;

5      Compute the estimated quantile indices $\{\boldsymbol{u}_r^{\boldsymbol{\theta}}\}_{1 \le r \le n}$ of the quantiles $\{f(\boldsymbol{x}_r)\}_{1 \le r \le n}$ w.r.t. data given by $\{f(T_{\boldsymbol{\theta}}(\boldsymbol{x}_j^+))\}_{1 \le j \le m}$;

6      Compute the MSE loss $\frac{1}{n}\sum_{r=1}^n \|\boldsymbol{u}_r - \boldsymbol{u}_r^{\boldsymbol{\theta}}\|_2^2$;

7      Update the parameters $\boldsymbol{\theta}$ using gradient descent;

8 **return** $\boldsymbol{\theta}$

---

## 3.3 Theoretical Guarantees

Algorithm 1 minimizes quantile loss aiming to match the marginal distributions. The class-conditional distributions of the features of de-corrupted data may not match with that of the class-conditional features of the training data. A simple counter-example would be to permute the labels of the class-conditionals. Fig 3 contains one such illustrative example where the class-conditionals fail to match for a valid solution to quantile-loss minimization. The dataset is given by two-moons in two dimensions with each moon denoting a class. If the test data is known to be a rotation of the train data about the origin. Observe that there exists two rotations on the test data that match the marginal distributions of the rotated test data and the train data - one solution is the correct operator, the other swaps the class-conditionals perfectly.

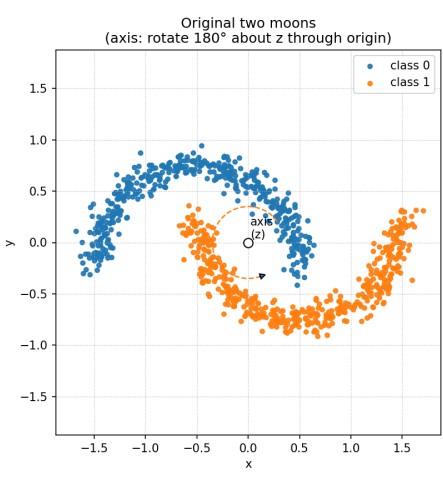

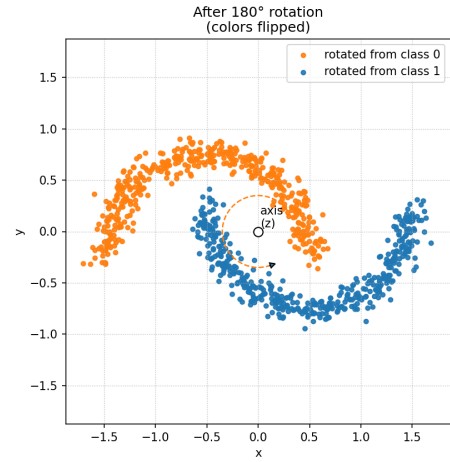

(a) The classes are given by the colors

(b) A 180 degree rotation of the data about the origin.

Figure 3: An illustrative example where marginal distributions align perfectly but the class-conditionals are flipped. If the de-corruption operator class were chosen from rotations about the origin, it is possible to obtain both the correct operator as well as the one that flips class-conditionals perfectly.

**Alignment upto Optimal Transport is Necessary but not Sufficient** Let $\mu_{\boldsymbol{X},Y}$, $\nu_{\boldsymbol{X},Y}$ and $\nu_{\boldsymbol{\theta},\boldsymbol{X},Y}$ denote the clean joint distribution, the corrupted joint distribution and the de-corrupted joint distribution at parameter $\boldsymbol{\theta}$ respectively. Let the features obtained from the clean and the de-corrupted distributions at parameter $\boldsymbol{\theta}$ using the frozen classifier be denoted by $\mu^f = f_{\#}\mu_{\boldsymbol{X}}$ and $\nu_{\boldsymbol{\theta}}^f = f_{\#}\nu_{\boldsymbol{X},\boldsymbol{\theta}}$ respectively. Even if feature marginals are matched exactly i.e. $\mu^f = \nu_{\boldsymbol{\theta}}^f$, classification risk on the denoised images can be strictly greater than the clean risk i.e. there exist distributions for which $\mu^f = \nu_{\boldsymbol{\theta}}^f$ but

$$\mathbb{P}_{(\boldsymbol{X}^+,Y)\sim\nu_{\boldsymbol{\theta},\boldsymbol{X},Y}}[\arg\max(h\circ f)(\boldsymbol{X}^+)\neq y|Y=y] > \mathbb{P}_{(\boldsymbol{X},Y)\sim\mu_{\boldsymbol{X},Y}}[\arg\max(h\circ f)(\boldsymbol{X})\neq y|Y=y] \qquad (10)$$

Fig 3 provides one such example.

Since we work with covariate shifts, we assume a joint distribution $\mu_{\boldsymbol{X}^+,\boldsymbol{X}}$ on the corrupt-clean pairs of images. Given a feature extractor $f$, one can work with the joint distribution $\mu_{\boldsymbol{X}^+,\boldsymbol{X}}^f$ i.e. $(f(\boldsymbol{X}^+), f(\boldsymbol{X}))$ where $(\boldsymbol{X}^+,\boldsymbol{X})\sim\mu_{\boldsymbol{X}^+,\boldsymbol{X}}$. A proxy for aligning the class-conditionals would be to minimize the mean-squared error (MSE) between the features of denoised corrupt image and its corresponding clean counterpart. However, in practice it is not possible to train based on minimizing MSE as paired samples cannot be obtained. Theorem 4 provides sufficient conditions under which minimizing the quantile loss is equivalent to minimizing the MSE. We shall require proposition 1 for the proof of Theorem 4.

**Proposition 1.** *(**Quantile Basis**) Let $F$ denote a probability distribution in $\mathbb{R}^d$ such that $\{x-y : x, y \sim F\}$ spans $\mathbb{R}^d$. Let $Supp(F)$ denote the support of $F$. Define the quantile index function $U_F : \mathbb{R}^d \to \mathbb{R}^d$ as*

$$U_F(\boldsymbol{z}) = E_{\boldsymbol{X}\sim F}\left[\frac{\boldsymbol{z}-\boldsymbol{X}}{\|\boldsymbol{z}-\boldsymbol{X}\|}\right] \qquad (11)$$

*then $U_F|_{Supp(F)}$ uniquely determines a probability distribution on $Supp(F)$.*

Informally, the result implies that if the support of the probability distribution is well-behaved in the underlying dimension then the quantiles outside of the support are redundant for uniquely identifying a probability distribution. The reader may refer to the appendix for a proof of Proposition 1 for a discrete probability distribution with finite support.

**Theorem 4.** *(**Equivalence of Quantile Loss and Pairwise Mean-Squared Error Minimization**) Let $\mu_{\boldsymbol{X},\boldsymbol{X}^+}$ denote the joint distribution on the covariate shift-original pairs of data in $\mathbb{R}^d \times \mathbb{R}^d$. Let $\mu_{\boldsymbol{X}^+}$ and*

$\mu_{\boldsymbol{X}}$ respectively denote the marginal distributions of covariate shifted and original data. Let $T_{\boldsymbol{\theta}} : \mathbb{R}^d \to \mathbb{R}^d$ with $\boldsymbol{\theta} \in \Theta$ denote the class of shift-correction operators. Let $\mu_{\boldsymbol{\theta}}$ denote the distribution of $T_{\boldsymbol{\theta}}(\boldsymbol{X}^+)$ where $\boldsymbol{X}^+ \sim \mu_{\boldsymbol{X}^+}$ Assume $\boldsymbol{X}^+, \boldsymbol{X}$ and $T_{\boldsymbol{\theta}}(\boldsymbol{X}^+)$ have finite second moments for all $\boldsymbol{\theta} \in \Theta$ and are absolutely continuous w.r.t. Lebesgue measure. For every $\boldsymbol{\theta} \in \Theta$, define the expected pairwise mean-squared error between the shift-corrected and the original data by

$$\mathcal{L}_{pair}(\boldsymbol{\theta}) = E_{\mu_{\boldsymbol{X}^+, \boldsymbol{X}}} \left[ \|T_{\boldsymbol{\theta}}(\boldsymbol{X}^+) - \boldsymbol{X}\|^2 \right] \tag{12}$$

Define the spatial quantile function $U_F : \mathbb{R}^d \to \mathbb{R}^d$ as

$$U_F(\boldsymbol{z}) = E_{\boldsymbol{X} \sim F} \left[ \frac{\boldsymbol{z} - \boldsymbol{X}}{\|\boldsymbol{z} - \boldsymbol{X}\|} \right] \tag{13}$$

Define quantile loss as

$$\mathcal{L}_{quant}(\boldsymbol{\theta}) = E_{\boldsymbol{Z} \sim \mu_{\boldsymbol{X}}} \left[ \|U_{\mu_{\boldsymbol{\theta}}}(\boldsymbol{Z}) - U_{\mu_{\boldsymbol{X}}}(\boldsymbol{Z})\|^2 \right] \tag{14}$$

Consider the following assumptions

- (A1) (**Regularity**) The span of the support $Supp(\mu_{\boldsymbol{X}})$ is $\mathbb{R}^d$ and $\boldsymbol{0} \in \mathbb{R}^d$ lies in the convex hull of $Supp(\mu_{\boldsymbol{X}})$.

- (A2a) (**Perfect Reconstruction**) $\exists \, \boldsymbol{\theta}^* \in \Theta$ such that $\mathcal{L}_{pair}(\boldsymbol{\theta}^*) = 0$.

- (A2b) (**Reconstruction**) $\boldsymbol{X} = T^*(\boldsymbol{X}^+) + \boldsymbol{\epsilon}$ with $E[\boldsymbol{\epsilon}|\boldsymbol{X}] = \boldsymbol{0}$ and $\|Cov(\boldsymbol{\epsilon}|\boldsymbol{X})\|_{op} \leq \delta$. Let $\inf_{\boldsymbol{\theta}} \mathcal{L}_{pair}(\boldsymbol{\theta}) = E[\|\boldsymbol{\epsilon}^2\|] = \sigma^2$ i.e. $\sigma^2 \leq d\delta$ denotes the irreducible error when trained to minimize pairwise mean-squared error.

- (A3a) (**Identifiability**) If $T_{\boldsymbol{\theta}_1}(\boldsymbol{X}^+) \stackrel{d}{=} T_{\boldsymbol{\theta}_2}(\boldsymbol{X}^+)$ for $\boldsymbol{\theta}_1, \boldsymbol{\theta}_2 \in \Theta$ then $T_{\boldsymbol{\theta}_1}(\boldsymbol{x}^+) = T_{\boldsymbol{\theta}_2}(\boldsymbol{x}^+)$ for $\mu_{\boldsymbol{X}^+}$-almost $\boldsymbol{x}^+$

- (A3b) (**Local Identifiability**) There exists $\theta_0$ and a small neighborhood around $\theta_0$ say $S_\epsilon(\theta_0) \subset \Theta$ such that if $T_{\boldsymbol{\theta}_1}(\boldsymbol{X}^+) \stackrel{d}{=} T_{\boldsymbol{\theta}_2}(\boldsymbol{X}^+)$ for $\boldsymbol{\theta}_1, \boldsymbol{\theta}_2 \in S_\epsilon(\theta_0)$ then $T_{\boldsymbol{\theta}_1}(\boldsymbol{x}^+) = T_{\boldsymbol{\theta}_2}(\boldsymbol{x}^+)$ for $\mu_{\boldsymbol{X}^+}$-almost $\boldsymbol{x}^+$.

then under the assumptions (A1), (A2a), (A3a), the following are equivalent

- (a) $\mathcal{L}_{pair}(\boldsymbol{\theta}) = 0$

- (b) $\mathcal{L}_{quant}(\boldsymbol{\theta}) = 0$

- (c) $\mu_{\boldsymbol{\theta}} \stackrel{d}{=} \mu_{\boldsymbol{X}}$

Let $\eta \geq \sigma^2$. Similarly, under the assumptions (A1), (A2b), (A3b) the following statements are equivalent

- (d) $\mathcal{L}_{pair}(\boldsymbol{\theta}_l) \to \Delta_1(\eta)$ as $l \to \infty$ where $\Delta_1 : \mathbb{R}^+ \to \mathbb{R}^+$ is a continuous increasing function with $\Delta_1(\sigma^2) = \sigma^2$.

- (e) $\mathcal{L}_{quant}(\boldsymbol{\theta}_l) \to \Delta_2(\eta)$ as $l \to \infty$ where $\Delta_2 : \mathbb{R}^+ \to \mathbb{R}^+$ is a continuous increasing function. Additionally, $\Delta_2(0) = 0$ when $\sigma = 0$.

- (f) $W_2(\mu_{\boldsymbol{\theta}_l}, \mu_{\boldsymbol{X}}) \to \Delta_3(\eta)$ as $l \to \infty$ where $\Delta_3 : \mathbb{R}^+ \to \mathbb{R}^+$ is a continuous increasing function. Additionally, $\Delta_3(0) = 0$ when $\sigma = 0$. $W_2(.,.)$ represents the Wasserstein-2 distance between two probability distributions.

Observe that $\Delta_2(\eta)$ is minimized $\Rightarrow \eta \approx \sigma^2 \Rightarrow \Delta_3(\eta)$ is minimized. Similarly, $\Delta_2(\eta)$ is minimized $\Rightarrow \eta \approx \sigma^2 \Rightarrow \Delta_1(\eta) \approx \sigma^2$. Intuitively, the result implies that under an appropriate initialization, minimizing quantile loss minimizes the Wasserstein distance approximately aligning the marginal distributions as much as the capacity of the decorruptor architecture allows. Also, this is equivalent to minimizing the pairwise errors

implicitly learning the optimal adapter for the given decorruptor architecture. The reader may refer to the appendix for a sketch of proof.

Since we minimize the quantile loss in the feature-space, theorem 4 is applicable only when the three assumptions are valid for the feature extractor i.e. $f \circ T_{\boldsymbol{\theta}}$. (A1) is a valid assumption as typical neural network feature representations are not sparse and the number of training samples of $\mu_{\boldsymbol{X}}$ are much larger than the dimension of the feature space. Assumption (A2b) is valid for real covariate-shifts as constructed in Hendrycks & Dietterich (2019). Thanks to the non-trivial performance of the classifier on the corrupted images, (A3b) holds locally around the parameters of the identity operator $\boldsymbol{\theta}_0$.

Fig 4 provides empirical evidence to Theorem 4. The classifier is a ResNet18 trained on CIFAR10 training images and is frozen while training the decorruptor. The corrupt dataset is a level 5 Gaussian distortion of CIFAR10C. The decorruptor is trained by minimizing the quantile loss. The mean-squared error is computed by using the paired test images of CIFAR10. Fig 4a, a scatter plot between the two losses indicates that the MSE and quantile losses are positively correlated. In fig 4b, both the quantile and mean-squared error losses are tracked as a function of steps. The losses are normalized (rescaled by positive constants) to accommodate on the same scale. The plot shows that both the losses reduce as the training progresses.

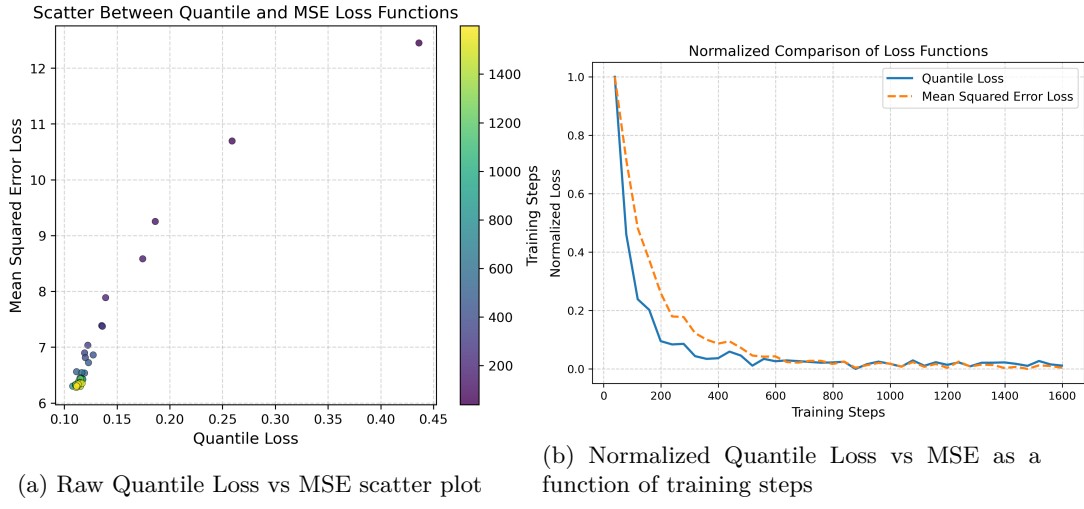

(a) Raw Quantile Loss vs MSE scatter plot

(b) Normalized Quantile Loss vs MSE as a function of training steps

Figure 4: The errors are plotted on the de-corrupted features of Gaussian distorted CIFAR10C images at severity level 5 (using ResNet18) as a function of epochs. The mean-squared error is computed using the test images of CIFAR10. Fig 4a is a scatter plot between raw quantile loss and means-squared error losses. In Fig 4b Quantile loss versus means-squared error loss is plotted as a function of epochs after normalizing them to accommodate on the same scale. Observe that the two losses are positively correlated and the losses reduce with the number of training steps.

## 4 Implementation Details

### 4.1 High-Confidence Uniform Sampling

Assuming regularity conditions, Proposition 1 implies matching the quantile indices of every vector in the support perfectly matches distributions. However, in practice using uninformative vectors would lead to sub-optimal solutions. For example, the quantiles chosen away from high-density regions of the data support can be omitted to boost the results at a lower computational cost. Fig 8 in the appendix provides a visual intuition.

Let $\mu^f_{HUS}$ denote a class-wise balanced high-confidence sampling distribution from the classifier features of the training distribution. Imposing the penalty on these quantiles, the quantile loss in Eq 9 changes to:

$$\mathcal{L}_{quant}(\boldsymbol{\theta}) = E_{\boldsymbol{Z} \sim \mu^f_{HUS}} \left[ \| U_{\nu^f_{\boldsymbol{\theta}}}(\boldsymbol{Z}) - U_{\mu^f}(\boldsymbol{Z}) \|^2 \right] \tag{15}$$

## 4.2 Efficient Model Scaling: Small-Batch Workarounds for Limited GPUs

One of the main reasons which made deep learning scalable is the fact that most of the loss functions are sample-separable. A loss function is sample-separable if the loss function decomposes into a sum of per-sample losses. A sample-separable loss enables the gradient of the total loss to be approximated with the expectation of gradients of per-mini-batch losses Nacson et al. (2019). Unfortunately, the quantile loss function (Eq 9) is not sample-separable. To circumvent this restriction we propose *composite-separability*.

**Definition 2.** *A loss function $L(\boldsymbol{x}_1, \cdots, \boldsymbol{x}_n; \boldsymbol{\theta})$ is said to be **composite-separable** if it can be decomposed into sum of sub-losses such that each sub-loss is a composition of a sample-separable function and another function which is possibly non-linear in that order i.e. there exists a decomposition such that*

$$L(\boldsymbol{x}_1, \cdots, \boldsymbol{x}_n; \boldsymbol{\theta}) = \frac{1}{|R|} \sum_{r \in R} g_r \left( \frac{1}{n} \sum_{i=1}^{n} h_r(\boldsymbol{x}_i; \boldsymbol{\theta}) \right) \tag{16}$$

*where $g_r$ is possibly non-linear and $h_r$ is a function of one variable for each $r \in R$*

A composite-separable loss would be sample-separable if each $g_r$ is linear. Conversely, every sample-separable loss function can be written in the form of Eq 16. Hence composite-separable loss functions naturally extend sample-separable loss functions.

Let $f_{\boldsymbol{\theta}}$ denote the neural-network. The quantile-loss defined by Eq 9 can be cast as a composite-separable loss as follows. $R \subsetneq \{1, 2, \cdots, n\}$ denotes the set of quantiles at which we match the indices. For each $r \in R$, $g_r : \mathbb{R}^k \to \mathbb{R}$ is given by $g_r(\boldsymbol{x}) = ||\boldsymbol{x} - \boldsymbol{u}_r||^2$ where $\boldsymbol{u}_r$ is the sample quantile index and $h_r : \mathbb{R}^k \to \mathbb{R}^k$ is given by

$$h_r(\boldsymbol{x}_i; \boldsymbol{\theta}) = \frac{\boldsymbol{Z}_r - f_{\boldsymbol{\theta}}(\boldsymbol{x}_i)}{||\boldsymbol{Z}_r - f_{\boldsymbol{\theta}}(\boldsymbol{x}_i)||} \tag{17}$$

Thanks to the structure of composite-separable loss functions, implementing SGD only requires obtaining a low-variance estimate of the average of the $h_r$ terms. This leverages the control-variate principle Ross (2022), which provides a systematic way to reduce the variance of stochastic estimators without altering their expectation. In our setting, this idea is realized by maintaining a memory bank on the CPU that stores the following quantities:

1. the feature vectors of all the inputs $\{f_{\boldsymbol{\theta}}(\boldsymbol{x}_i)\}$ at $\boldsymbol{\theta} = \boldsymbol{\theta}_t$. At each gradient step, the feature vectors of the indices corresponding to the batch are updated.

2. the average of $h_r$-functions over all samples for each $r$ i.e. $\frac{1}{n} \sum_{i=1}^{n} h_r(\boldsymbol{x}_i; \boldsymbol{\theta})$ at $\boldsymbol{\theta} = \boldsymbol{\theta}_{t_{snap}}$. At the end of each epoch, $t_{snap}$ is reset to $t$ i.e. the averages of $h_r$-functions are updated with the current parameters.

Let $\{\boldsymbol{\theta}_t\}$ denote the time-series of the parameter updates. Assume for simplicity that the batch size divides the total number of samples. Let $B$ denote a batch, $b$ denote the batch size and $g$ denote the number of gradient steps per epoch then number of samples $n = gb$.

The estimate of $\frac{1}{n} \sum_{i=1}^{n} h_r(\boldsymbol{x}_i; \boldsymbol{\theta})$ at each gradient step $t$ is given by

$$\frac{1}{b} \sum_{i \in B} h_r(\boldsymbol{x}_i; \boldsymbol{\theta}_t) + \left( \frac{1}{n} \sum_{i=1}^{n} h_r(\boldsymbol{x}_i; \boldsymbol{\theta}_{t_{snap}}) - \frac{1}{b} \sum_{i \in B} h_r(\boldsymbol{x}_i; \boldsymbol{\theta}_{t_{snap}}) \right) \tag{18}$$

The rate at which the feature vectors and the corresponding $h_r$ values in the memory bank are updated determines that $t - t_{snap} \leq g$.

Under a sufficiently small learning rate, frequent updates to the memory bank ensure that $\frac{1}{b} \sum_{i \in B} h_r(\boldsymbol{x}_i; \boldsymbol{\theta}_t)$ and $\frac{1}{b} \sum_{i \in B} h_r(\boldsymbol{x}_i; \boldsymbol{\theta}_{t_{snap}})$ remain highly and positively correlated. Intuitively, the negative sign in the

snapshot-based correction term drives the two batch averages toward a common mean, thereby canceling a substantial portion of their shared variability and reducing the overall gradient variance. This mechanism is closely related to the control logic underlying Stochastic Variance Reduced Gradient (SVRG) Johnson & Zhang (2013).

The same intuition applies to any composite-separable loss. Notably, quantile loss arises as a particular instance of such losses, exhibiting second-order interactions among samples. A detailed analysis of the memory-bank estimator for quantile loss is provided in appendix A.2.

## 5 Experiments

For experimental validation, we chose three datasets CIFAR10C, CIFAR100C and TINYIMAGENETC with number of classes ranging from 10 to 200. We run all the experiments on corruption level 5. We use ResNet18 as a representative for classic convolution-based networks and three transformer networks suited for $32 \times 32 \times 3$ images namely compact convolution transformer (CCT), compact vision transformer (CVT) and a lightweight vision transformer (ViT-Lite) Hassani et al. (2021) to validate on transformer networks. The code is available at `https://anonymous.4open.science/r/GeometricQuantile-TTA-BE02/README.md`

Each of these networks are trained on the training datasets of CIFAR10, CIFAR100 and TINYIMAGENET. The ResNet models for all datasets are trained using cross-entropy loss. The CCT models for CIFAR10 and CIFAR100 are pre-trained SSL models whose weights are obtained from the official repository Hassani et al. (2021). The CVT, ViT-Lite models for all datasets and CCT for TinyImageNet are trained using cross-entropy loss as pre-trained SSL weights are unavailable and SSL requires expensive training. Also, regularization terms to penalize the deviation of the batch statistics of channel means and standard deviations of the decorrupted images from standard values are added to the quantile loss in the implementation. The decorruptor is a convolutional network with spatial convolutions of various sizes and a deconvolution containing 6.2M parameters.

### 5.1 Results

Table 1: Increase in test accuracy by the Quantile-approach compared to the baseline. Baselines are obtained by applying the pre-trained classifier directly to the corrupted datasets. Current SOTA are the accuracies obtained by SoTTA Gong et al. (2024) whenever applicable

| Dataset (# classes) | Model | Baseline | Quantile | Increase | Current SOTA |
|---|---|---|---|---|---|
| CIFAR10C (10) | ResNet18 | 54.4% | 79.4% | +25.0% | |
| | CCT | 80.3% | **88.8%** | +8.5% | 82.2%* |
| | CVT | 54.6% | 70.3% | +15.7% | |
| | ViT-Lite | 53.2% | 69.0% | +15.8% | |
| CIFAR100C (100) | ResNet18 | 38.7% | 49.9% | +11.2% | |
| | CCT | 58.8% | **65.0%** | +6.2% | 60.5%† |
| | CVT | 40.6% | 46.5% | +5.9% | |
| | ViT-Lite | 39.2% | 43.1% | +3.9% | |
| TINYIMAGENETC (200) | ResNet18 | 20.8% | 25.4% | +4.6% | |
| | CCT | 12.8% | 16.0% | +3.2% | NA** |
| | CVT | 12.5% | 16.1% | +3.6% | |
| | ViT-Lite | 12.3% | 15.8% | +3.5% | |

Table 1 provides average improvement across 15 types of corruptions obtained on each of the datasets by the quantile approach compared to directly applying the classifier to the corrupted dataset. Baseline model refers to using the classifier trained on clean counter-parts i.e. CIFAR10, CIFAR100 and TINYIMAGENET directly on CIFAR10C, CIFAR100C and TINYIMAGENETC. Current SOTA corresponds to the accuracy

obtained by SoTTA Gong et al. (2024) and are taken directly from the article. Although our ResNet18-based results fall short of SoTTA on both CIFAR10C and CIFAR100C (79.4% vs. 82.2% and 49.5% vs. 60.5%, respectively), the strength of our approach lies in its architecture independence. Unlike SoTTA, which is tailored to CNNs, our method applies seamlessly to transformer architectures which are widely known to surpass CNN-based classifiers in performance. This flexibility is not merely conceptual: when instantiated on CCT, our method achieves 65.0% on CIFAR100C, outperforming SoTTA's ResNet18 result of 60.5%. *The increase of 6.6% accuracy on CIFAR10C and 4.5% on CIFAR100C compared to the current SOTA is attributed to the applicability of quantile-loss minimization to transformer architectures*[1]. Thus, while there is a slight performance gap on CNNs, our method compensates for it through broader applicability and demonstrably stronger performance on more advanced architectures, underscoring its long-term potential.

Table 2: Drop in test accuracy of the Quantile-approach and the current SOTA (SoTTA Gong et al. (2024)) w.r.t. the clean counterparts

| Approach-Architecture | CIFAR10 | CIFAR10C | Drop | CIFAR100 | CIFAR100C | Drop |
|---|---|---|---|---|---|---|
| SoTTA-ResNet18 | 95.2% | 82.2% | 13.0% | 78.6% | 60.5% | 18.1% |
| Quantile-ResNet18 | 92.0% | 79.9% | 12.1% | 72.5% | 49.9% | 23.1% |
| Quantile-CCT | 96.2% | 88.8% | **7.4%** | 80.9% | 65.0% | **15.9%** |

Table 2 provides the comparison of the average accuracy drop of the ResNet18 models on the corrupt test data (CIFAR10C and CIFAR100C) w.r.t. the accuracies on the clean test data counterparts i.e. CIFAR10 and CIFAR100 respectively. The results of SoTTA are tabulated from the article Gong et al. (2024). We trained the ResNet18 from scratch and obtained the accuracies on CIFAR10 and CIFAR100 as tabulated in the second row. The pre-trained CCT weights are used to generate the results for the third row. Observe that the drops in accuracies are comparable in all the cases. On the other hand, quantile approach outperforming the current SOTA is purely attributed to the fact that they are applicable to transformers but the current SOTA are not.

## 5.2 Runtime and Memory Cost Analysis

Quantile loss minimization enables learning a decorruptor without paired clean-corrupt images. Our implementation adds two components on top of a frozen classifier: (i) quantile loss with a memory bank, and (ii) a decorruptor network. Recall that the memory bank stores, at each optimization step: (i) source features, (ii) decorrupted features, and (iii) quantile indices of reference quantiles computed with respect to the empirical distributions of both the source and decorrupted features (the latter evaluated at the snapshot parameter). These cached quantities enable low-variance quantile-loss estimation without recomputing distribution statistics at each step as discussed in Subsection 4.2

A typical set-up in our experiments is a CIFAR100C with ResNet18 ($10,000$ source features with 512 dimension, batch size 128), the quantile-loss module needs $\approx 570$ MB and $\approx 2$ ms per batch on an Nvidia H100. Unlike mean-teacher TTA methods (applicable to transformers) which duplicate the classifier and thus scale with classifier complexity, our overhead scales only with the complexity of corruption (i.e. the decorruptor architecture). In our experiments, the 6.2 M parameter decorruptor increases training time from 1.44 s to 2.03 s per epoch and peak memory from 765 MB to 1429 MB which is comparable to mean-teacher-based methods, yet more principled, because it remains independent of the classifier architecture.

## 5.3 Dependence on Hyperparameters

Quantile loss minimization involves choice of two hyperparameters - the number of quantiles to choose (i.e. support of $\mu_{HUS}^f$ in Eq 9) and the batch-size. We ran experiments with batch sizes of 128 and 256 respectively

---

[1]* †** The SOTA accuracy refers to accuracy of current SOTA on models with parameters at most that of ResNet18. SOTA is not applicable to CCT, CVT and ViT-Lite. The results on TinyImageNetC are not readily available in the article. Hence the results are not tabulated.

on CIFAR100C dataset using the pre-trained CCT transformer model. For each of the batch sizes, we vary the the number of quantiles to 1000 and 2000. These results are compared to that of the results provided in table 1 which uses 1000 quantiles and 512 batch size. The results in table 3 across batch sizes and number of quantiles are similar indicating that the algorithm scales well with larger models that force smaller batch sizes. Also, approximately 10 quantiles per class suffice assuming the classifier is well-trained on the training distribution allowing for well-separable clusters. Intuitively, the hyperparameter - number of quantiles can be set as an estimated maximum number of clusters per class-conditional multiplied by number of classes.

Table 3: Test accuracy obtained by the CCT Quantile on CIFAR100 dataset with small batch sizes and varying number of quantiles.

| Dataset (# classes) | Batch Size | # Quantiles per Class | Accuracy |
|---|---|---|---|
| CIFAR100C (100) | 128 | 10 | 65.5% |
| | 256 | 10 | 65.8% |
| | 128 | 20 | 66.8% |
| | 256 | 20 | 65.6% |
| | 512 | 10 | 65.0% |

## 6 Conclusions and Perspectives

In this article, we propose an architecture-agnostic design to solve the TTA problem by explicitly learning a decorruptor from the test distribution to the train distribution. The solution is based on matching the classifier-feature distributions of the train and the decorrupted images. We design a novel loss function based on geometric quantiles to quantify closeness between high-dimensional distributions and provide an efficient algorithm to implement the loss function. We experimentally show that it works on both CNN-based and transformer-based classifiers. Further, we provide a theoretical result showing the existence of 'good' initialization of the network such that the quantile loss minimization is theoretically equivalent to a training regime based on paired corrupt and clean images in principle. Thus, quantile loss minimization is shown to effectively align class-conditionals without the need of class labels assuming 'good' initialization.

Quantile loss minimization algorithm efficiently matches high-dimensional probability distributions. It would enable scalable optimal transport–based domain adaptation, and improved self-supervised representation learning through tighter matching of augmented view statistics. In generative modeling, efficient distribution matching could stabilize adversarial training, accelerate convergence of diffusion and flow-based models, and provide more reliable likelihood-free inference. On the theoretical side, characterizing the quantile basis of a probability distribution etc would allow us to design more efficient algorithms.

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

Table 4: CIFAR10C - Comparison of Accuracies across noise-types at severity level 5.

| | Gaussian | Shot | Impulse | Defocus | Glass | Motion | Zoom | Snow | Frost | Fog | Brightness | Contrast | Elastic | Pixelate | Jpeg | Mean |
|---|---|---|---|---|---|---|---|---|---|---|---|---|---|---|---|---|
| ResNet18 Source | 36.4 | 43.3 | 28.0 | 48.4 | 44.7 | 55.8 | 58.4 | 70.6 | 61.6 | 61.1 | 86.0 | 23.4 | 72.3 | 50.5 | 75.2 | 54.4 |
| ResNet18 Quantile | 76.9 | 80.3 | 79.5 | 81.6 | 68.1 | 80.5 | 80.8 | 80.8 | 79.6 | 79.5 | 86.8 | 80.2 | 75.0 | 81.5 | 79.6 | 79.4 |
| Increase | +40.5 | +37.0 | +51.5 | +33.2 | +23.4 | +24.7 | +22.4 | +10.2 | +18.0 | +18.4 | +0.8 | +56.8 | +2.7 | +31.0 | +4.4 | +25.0 |
| ResNet18 SoTTA | 75.0 | 77.5 | 68.8 | 88.8 | 70.7 | 87.5 | 89.0 | 85.4 | 84.0 | 88.2 | 91.9 | 83.9 | 79.8 | 83.9 | 78.3 | 82.2 |
| CCT Source | 63.1 | 67.6 | 42.3 | 92.4 | 74.0 | 89.7 | 92.5 | 89.6 | 88.7 | 85.5 | 95.1 | 88.7 | 86.2 | 64.8 | 84.4 | 80.3 |
| CCT Quantile | 86.7 | 87.9 | 87.6 | 90.8 | 82.6 | 89.9 | 90.7 | 90.1 | 90.3 | 88.6 | 93.4 | 91.0 | 85.9 | 90.2 | 86.5 | 88.8 |
| Increase | +23.6 | +20.3 | +45.3 | -1.6 | +8.6 | +0.2 | -1.8 | +0.5 | +1.6 | +3.1 | -1.7 | +2.3 | -0.3 | +25.4 | +2.1 | +8.5 |
| CVT Source | 35.4 | 39.5 | 29.4 | 56.2 | 59.8 | 60.2 | 62.5 | 66.8 | 59.2 | 42.8 | 75.3 | 20.7 | 74.0 | 68.0 | 69.0 | 54.6 |
| CVT Quantile | 70.1 | 72.1 | 72.6 | 75.1 | 67.9 | 73.4 | 74.6 | 74.3 | 71.8 | 65.3 | 76.2 | 42.9 | 72.1 | 76.2 | 69.8 | 70.3 |
| Increase | +34.7 | +32.6 | +43.2 | +18.9 | +8.1 | +10.3 | +7.2 | +7.5 | +12.6 | +22.5 | +0.9 | +22.2 | -1.9 | +8.2 | +0.8 | +15.7 |
| ViT-Lite Source | 32.7 | 37.0 | 30.7 | 53.0 | 59.9 | 57.9 | 59.7 | 64.4 | 55.2 | 41.4 | 73.3 | 23.5 | 72.9 | 67.5 | 68.9 | 53.2 |
| ViT-Lite Quantile | 67.7 | 71.0 | 71.5 | 73.1 | 66.8 | 74.2 | 75.1 | 72.5 | 71.3 | 63.7 | 75.6 | 37.3 | 71.3 | 74.7 | 68.7 | 69.0 |
| Increase | +35.0 | +34.0 | +40.8 | +20.1 | +6.9 | +16.3 | +15.4 | +8.1 | +16.1 | +22.3 | +2.3 | +13.8 | -1.6 | +7.2 | -0.2 | +15.8 |

Table 5: CIFAR100C - Comparison of Accuracies across noise-types at severity level 5.

| | Gaussian | Shot | Impulse | Defocus | Glass | Motion | Zoom | Snow | Frost | Fog | Brightness | Contrast | Elastic | Pixelate | Jpeg | Mean |
|---|---|---|---|---|---|---|---|---|---|---|---|---|---|---|---|---|
| ResNet18 Source | 12.3 | 15.8 | 22.8 | 50.5 | 21.8 | 46.1 | 51.2 | 47.3 | 36.3 | 43.4 | 69.9 | 59.8 | 45.9 | 15.2 | 41.7 | 38.7 |
| ResNet18 Quantile | 42.1 | 45.5 | 40.9 | 57.4 | 34.3 | 56.4 | 54.0 | 55.0 | 50.4 | 46.5 | 68.4 | 54.9 | 42.0 | 53.4 | 47.6 | 49.9 |
| Increase | +29.8 | +29.7 | +18.1 | +6.9 | +12.5 | +10.3 | +2.8 | +7.7 | +14.1 | +3.1 | -1.5 | -4.9 | -3.9 | +38.2 | +5.9 | +11.2 |
| ResNet18 SoTTA | 52.0 | 53.4 | 45.0 | 68.8 | 49.1 | 66.7 | 69.0 | 61.7 | 60.2 | 64.7 | 72.2 | 66.4 | 58.6 | 64.1 | 55.0 | 60.5 |
| CCT Source | 42.5 | 45.1 | 42.4 | 73.1 | 43.6 | 69.7 | 73.0 | 67.5 | 64.0 | 55.3 | 76.3 | 63.1 | 64.0 | 42.8 | 59.2 | 58.8 |
| CCT Quantile | 61.6 | 62.5 | 66.3 | 71.9 | 55.0 | 70.1 | 68.2 | 68.5 | 67.9 | 60.3 | 75.9 | 56.3 | 59.5 | 69.4 | 60.8 | 65.0 |
| Increase | +19.1 | +17.4 | +23.9 | -1.2 | +11.4 | +0.4 | -4.8 | +1.0 | +3.9 | +5.0 | -0.4 | -6.8 | -4.5 | +26.6 | +1.6 | +6.2 |
| CVT Source | 15.8 | 18.7 | 11.8 | 51.2 | 31.6 | 50.9 | 54.9 | 47.9 | 41.0 | 38.7 | 63.8 | 43.0 | 53.9 | 42.6 | 43.2 | 40.6 |
| CVT Quantile | 34.8 | 40.1 | 32.0 | 56.6 | 41.3 | 54.4 | 53.2 | 52.9 | 50.9 | 35.0 | 62.3 | 28.1 | 48.6 | 59.0 | 47.8 | 46.5 |
| Increase | +19.0 | +21.6 | +20.2 | +5.4 | +9.7 | +3.5 | -1.7 | +5.0 | +9.9 | -3.7 | -1.5 | -14.9 | -5.3 | +16.4 | +4.6 | +5.9 |
| ViT-Lite Source | 17.9 | 19.8 | 12.8 | 48.8 | 31.1 | 47.7 | 51.9 | 45.3 | 38.2 | 35.9 | 60.7 | 40.0 | 51.4 | 43.9 | 43.2 | 39.2 |
| ViT-Lite Quantile | 34.5 | 36.1 | 29.5 | 53.9 | 40.4 | 51.4 | 50.1 | 49.1 | 47.2 | 29.9 | 60.8 | 22.8 | 42.4 | 54.3 | 43.5 | 43.1 |
| Increase | +16.6 | +16.3 | +16.7 | +5.1 | +9.3 | +3.7 | -1.8 | +3.8 | +9.0 | -6.0 | +0.1 | -17.2 | -9.0 | +10.4 | +0.3 | +3.9 |

# A  Appendix

## A.1  Detailed Experimental Results and Proofs

**Lemma 5.** *Suppose $S = \{x_1, \cdots, x_n\}$ span $\mathbb{R}^d$. Let $D = \{x_i - x_j : 1 \leq i, j \leq n\}$, $D_j = \{x_i - x_j : 1 \leq i \leq n\}$ for every $1 \leq j \leq n$. If the zero vector $\mathbf{0} \in ConvHull(S)$ then*

1. *$D$ spans $\mathbb{R}^d$, and*

2. *$D_j$ spans $\mathbb{R}^d$ for every $1 \leq j \leq n$*

*Proof.* It is easy to see that for any $1 \leq j \leq n$ $Span(D_j) \subseteq Span(D) \subseteq Span(S)$ as $D_j \subset D$ and $D \subset Span(S)$. To prove the first part, it is enough to show that $S \subset Span(D)$. Since $\mathbf{0} \in ConvHull(S)$, there exists $0 \leq \lambda_k \leq 1$ such that

$$\sum_{k=1}^{n} \lambda_k x_k = \mathbf{0} \tag{19}$$

Table 6: TINYIMAGENETC - Comparison of Accuracies across noise-types at severity level 5.

| | Gaussian | Shot | Impulse | Defocus | Glass | Motion | Zoom | Snow | Frost | Fog | Brightness | Contrast | Elastic | Pixelate | Jpeg | Mean |
|---|---|---|---|---|---|---|---|---|---|---|---|---|---|---|---|---|
| ResNet18 Source | 8.3 | 11.0 | 7.0 | 13.7 | 10.1 | 24.5 | 25.1 | 25.0 | 30.5 | 13.3 | 33.0 | 9.1 | 30.7 | 28.2 | 42.7 | 20.8 |
| ResNet18 Quantile | 15.6 | 18.1 | 7.9 | 22.1 | 10.1 | 32.4 | 33.0 | 33.6 | 29.5 | 20.7 | 38.2 | 3.6 | 31.9 | 43.5 | 40.5 | 25.4 |
| Increase | +7.3 | +7.1 | +0.9 | +8.4 | +0.0 | +7.9 | +7.9 | +8.6 | -1.0 | +7.4 | +5.2 | -5.5 | +1.2 | +15.3 | -2.2 | +4.6 |
| CCT Source | 3.9 | 4.6 | 2.9 | 12.5 | 9.5 | 18.8 | 19.6 | 14.6 | 17.5 | 8.1 | 15.1 | 2.5 | 23.7 | 8.2 | 31.1 | 12.8 |
| CCT Quantile | 8.1 | 9.7 | 6.9 | 15.1 | 9.7 | 20.0 | 21.5 | 20.2 | 15.5 | 10.4 | 20.6 | 2.0 | 24.3 | 27.4 | 27.8 | 16.0 |
| Increase | +4.2 | +5.1 | +4.0 | +2.6 | +0.2 | +1.2 | +1.9 | +5.6 | -2.0 | +2.3 | +5.5 | -0.5 | +0.6 | +19.2 | -3.3 | +3.2 |
| CVT Source | 3.6 | 4.9 | 3.0 | 13.5 | 12.6 | 20.3 | 19.8 | 12.3 | 17.6 | 6.1 | 14.2 | 3.1 | 24.5 | 6.3 | 25.9 | 12.5 |
| CVT Quantile | 8.0 | 10.1 | 7.2 | 15.9 | 11.4 | 23.6 | 22.6 | 19.7 | 16.8 | 8.0 | 20.1 | 1.9 | 24.1 | 26.3 | 25.6 | 16.1 |
| Increase | +4.4 | +5.2 | +4.2 | +2.4 | -1.2 | +3.3 | +2.8 | +7.4 | -0.8 | +1.9 | +5.9 | -1.2 | -0.4 | +20.0 | -0.3 | +3.6 |
| ViT-Lite Source | 4.1 | 5.5 | 3.0 | 13.6 | 12.2 | 19.4 | 19.9 | 11.9 | 16.2 | 6.3 | 12.8 | 2.8 | 23.7 | 6.6 | 26.0 | 12.3 |
| ViT-Lite Quantile | 10.0 | 13.0 | 9.6 | 13.8 | 10.3 | 21.0 | 22.2 | 17.4 | 14.9 | 7.2 | 21.2 | 1.9 | 24.3 | 25.0 | 24.7 | 15.8 |
| Increase | +5.9 | +7.5 | +6.6 | +0.2 | -1.9 | +1.6 | +2.3 | +5.5 | -1.3 | +0.9 | +8.4 | -0.9 | +0.6 | +18.4 | -1.3 | +3.5 |

Table 7: CIFAR100C - Comparison of CCT-Quantile Accuracies at severity level 5 across varying batch sizes and number of quantiles.

| | Gaussian | Shot | Impulse | Defocus | Glass | Motion | Zoom | Snow | Frost | Fog | Brightness | Contrast | Elastic | Pixelate | Jpeg | Mean |
|---|---|---|---|---|---|---|---|---|---|---|---|---|---|---|---|---|
| B=128, Q=1000 | 62.3 | 63.0 | 67.8 | 73.8 | 56.9 | 72.0 | 69.9 | 70.1 | 68.7 | 61.2 | 76.3 | 51.6 | 60.0 | 68.6 | 60.1 | 65.5 |
| B=256, Q=1000 | 62.2 | 63.6 | 67.9 | 72.1 | 56.6 | 70.2 | 67.8 | 68.8 | 69.2 | 60.3 | 77.1 | 57.7 | 61.1 | 70.1 | 62.1 | 65.8 |
| B=128, Q=2000 | 63.8 | 64.3 | 68.0 | 74.4 | 59.1 | 72.8 | 71.4 | 71.0 | 70.0 | 62.5 | 76.4 | 54.4 | 62.1 | 70.4 | 61.3 | 66.8 |
| B=256, Q=2000 | 63.3 | 64.3 | 68.1 | 71.5 | 55.2 | 70.9 | 67.9 | 67.8 | 67.0 | 60.8 | 75.5 | 57.8 | 62.1 | 70.0 | 62.3 | 65.6 |
| B=512, Q=1000 | 61.6 | 62.5 | 66.3 | 71.9 | 55.0 | 70.1 | 68.2 | 68.5 | 67.9 | 60.3 | 75.9 | 56.3 | 59.5 | 69.4 | 60.8 | 65.0 |

and

$$\sum_{k=1}^{n} \lambda_k = 1 \tag{20}$$

Any vector $x_i$ can be written as

$$x_i = x_i - \sum_{k=1}^{n} \lambda_k x_k = \sum_{k=1}^{n} \lambda_k (x_i - x_k) \in Span(D) \tag{21}$$

To complete the second part of the proof, observe that every vector in $D$ is in the span of $D_k$ because

$$x_i - x_j = (x_i - x_k) - (x_j - x_k) \tag{22}$$

$\square$

**Proof of Proposition 1**

*Proof.* We will present a proof by contradiction for a discrete probability distribution. Suppose $F$ is a discrete probability distribution with finite support say $n$ points $x_1, \cdots, x_n$. Define unit vectors for every $i \neq j$ and $1 \leq i, j \leq n$

$$v_{j,i} = \frac{x_j - x_i}{\|x_j - x_i\|} \tag{23}$$

Suppose $\sum_{i=1}^{n} p_i = 1$ and $\sum_{i=1}^{n} q_i = 1$ where $p_i > 0$ and $q_i > 0$ for all $1 \leq i \leq n$. Define $\delta_i := p_i - q_i$. It is enough to show that if

$$\sum_{i \neq j} \delta_i v_{j,i} = 0 \tag{24}$$

for all $1 \leq j \leq n$ then $\delta_i = 0$ for all $1 \leq i \leq n$.

Observe that Eq 24 can be compactly written as

$$M\boldsymbol{\delta} = 0 \tag{25}$$

where $M$ is a $nd \times n$ matrix and $\boldsymbol{\delta} = (\delta_1, \cdots, \delta_n)^T$ and also we have $\mathbf{1}^T \boldsymbol{\delta} = 0$. We need to show that the only solution to this system is $\boldsymbol{\delta} = \mathbf{0}$.

Lemma 5 and the given conditions imply that for any fixed index $i$, $\{x_j - x_i : j \neq i\}$ spans $\mathbb{R}^d$. For each $i$, there exists $1 \leq j_1, \cdots, j_d \leq n$ such that the $d$ vectors $v_{j_1,i}, \cdots, v_{j_d,i}$ are linearly independent. For each $1 \leq m \leq d$, consider the Eq 24 at $j = j_m$ i.e.

$$\sum_{l \neq j_m} \delta_l v_{j_m,l} = 0 \tag{26}$$

Combining them all we have

$$\begin{bmatrix} v_{j_1,1} & v_{j_1,2} & \cdots & v_{j_1,n} \\ v_{j_2,1} & v_{j_2,2} & \cdots & v_{j_2,n} \\ \vdots & \vdots & \ddots & \vdots \\ v_{j_d,1} & v_{j_d,2} & \cdots & v_{j_d,n} \end{bmatrix} \boldsymbol{\delta} = \mathbf{0} \tag{27}$$

Column $i$ has independent vectors hence $\delta_i = 0$. Since the fixed index $i$ was arbitrarily chosen, $\boldsymbol{\delta} = \mathbf{0}$. The proof for a general probability distribution is similar and requires more rigor. □

**Proof of theorem 4**

*Proof.* (a) $\Rightarrow$ (c) follows from the non-negativity of the squared-norm and the expectation over the support of the joint distribution.

(c) $\Rightarrow$ (a): $\mu_{\boldsymbol{\theta}} \stackrel{d}{=} \mu_{\boldsymbol{X}}$ and (A1) i.e. existence of a perfect reconstruction implies that $\mu_{\boldsymbol{\theta}*} \stackrel{d}{=} \mu_{\boldsymbol{X}}$ as we already proved (a) $\Rightarrow$ (c). Identifiability (A2) $\Rightarrow T_{\boldsymbol{\theta}}(\boldsymbol{x}^+) = T_{\boldsymbol{\theta}*}(\boldsymbol{x}^+)$ for $\mu_{\boldsymbol{X}^+}$-almost $\boldsymbol{x}^+$. Hence, $\mathcal{L}_{pair}(\boldsymbol{\theta}) = 0$

(c) $\Rightarrow$ (b) follows directly from the fact that $U_{\mu_{\boldsymbol{\theta}}}(\boldsymbol{z}) = U_{\mu_{\boldsymbol{X}}}(\boldsymbol{z})$ for every $\boldsymbol{z}$ in the common support of the distributions.

(b) $\Rightarrow$ (c) follows from (A3), Lemma 5 and Proposition 1.

The proofs of (d) $\Rightarrow$ (f), (f) $\Rightarrow$ (d), (e) $\Rightarrow$ (f), and (f) $\Rightarrow$ (e) are similar but require more rigor.

□

### A.2 Composite-Separable Functions

Some examples of well-known loss functions can be written as a composite-separable loss as follows:

1. **Mean-squared Loss** $|R| = 1$, $g_r$ is the identity mapping. $h_r(\boldsymbol{x}_i; \boldsymbol{\theta}) = (f_{\boldsymbol{\theta}}(\boldsymbol{x}_i) - \boldsymbol{y}_i)^2$.

2. **Cross-Entropy Loss** $|R| = 1$, $g_r$ is the identity mapping. $h_r(\boldsymbol{x}_i; \boldsymbol{\theta}) = -\boldsymbol{y}_i^T log\left(\frac{exp(f_{\boldsymbol{\theta}}(\boldsymbol{x}_i))}{\mathbf{1}^T exp(f_{\boldsymbol{\theta}}(\boldsymbol{x}_i))}\right)$. Here $log$ refers to element-wise application of logarithm, $\boldsymbol{y}_i$ is the one-hot vector of class labels, $\mathbf{1}$ refers to the vector with all ones.

**Justification of Memory Bank as a variance-reduced estimator**

**Lemma 6.** *Suppose $B$ is a batch of size $b$ drawn from a population of size $n$ say $\{\boldsymbol{x}_1, \cdots, \boldsymbol{x}_n\}$ without replacement then the variance of the sample mean (measured by expected squared norm)*

$$\bar{\boldsymbol{x}}_B = \frac{1}{b} \sum_{i \in B} \boldsymbol{x}_i \tag{28}$$

*is given by*

$$Var(\bar{\boldsymbol{x}}_B) := E\left[\|\bar{\boldsymbol{x}}_B - \boldsymbol{\mu}\|^2\right] = \frac{1}{b} \frac{n-b}{n-1} tr(\Sigma) \tag{29}$$

*where $\boldsymbol{\mu} = \frac{1}{n} \sum_{i=1}^{n} \boldsymbol{x}_i$ and*

$$\Sigma = \frac{1}{n} \sum_{i=1}^{n} (\boldsymbol{x}_i - \boldsymbol{\mu})(\boldsymbol{x}_i - \boldsymbol{\mu})^{\mathsf{T}} \tag{30}$$

*Proof.* For uniform sampling without replacement we have probability of selecting sample $i$ as $P\{I_i = 1\} = \frac{b}{n}$ and probability of selecting two samples $i$ and $j$ as $P\{I_i = 1, I_j = 1\} = \frac{\binom{n-2}{b-2}}{\binom{n}{b}}$.

$$\bar{\boldsymbol{x}}_B = \frac{1}{b} \sum_{i=1}^{n} I_i \boldsymbol{x}_i \tag{31}$$

The rest of the proof is straightforward as it involves calculation of variance and covariance terms using the marginal and joint probabilities. □

*Proof.* We will drop the subscript $r$ and denote $h_r(\boldsymbol{x}_i; \boldsymbol{\theta})$ with $h(\boldsymbol{x}_i; \boldsymbol{\theta})$. Note that each $h(\boldsymbol{x}_i; \boldsymbol{\theta})$ is a d-dimensional unit vector. The goal is to construct low variance estimators of

$$\boldsymbol{\mu}_t = \frac{1}{n} \sum_{i=1}^{n} h(\boldsymbol{x}_i; \boldsymbol{\theta}_t) \tag{32}$$

using mini-batches of indices from $\{1, \cdots, n\}$. We have two estimators at hand. The crude mini-batch estimator is given by

$$\hat{\boldsymbol{\mu}}_t^{crude} = \frac{1}{b} \sum_{i \in B} h(\boldsymbol{x}_i; \boldsymbol{\theta}_t) \tag{33}$$

Denote

$$\boldsymbol{\mu}_{snap} = \frac{1}{n} \sum_{i=1}^{n} h(\boldsymbol{x}_i; \boldsymbol{\theta}_{snap}) \tag{34}$$

The memory-bank based estimator with a constant $\beta$ is given by (we will substitute $\beta = 1$ later)

$$\hat{\boldsymbol{\mu}}_t^{control} = \frac{1}{b} \sum_{i \in B} h(\boldsymbol{x}_i; \boldsymbol{\theta}_t) + \beta \left( \boldsymbol{\mu}_{snap} - \frac{1}{b} \sum_{i \in B} h(\boldsymbol{x}_i; \boldsymbol{\theta}_{snap}) \right) \tag{35}$$

Both are unbiased estimators of $\boldsymbol{\mu}_t$

$$E_B \left[ \frac{1}{b} \sum_{i \in B} h(\boldsymbol{x}_i; \boldsymbol{\theta}_t) \right] = \frac{1}{n} \sum_{i=1}^{n} h(\boldsymbol{x}_i; \boldsymbol{\theta}_t) = \boldsymbol{\mu}_t \tag{36}$$

With a similar argument

$$E_B \left[ \hat{\boldsymbol{\mu}}_t^{control} \right] = \boldsymbol{\mu}_t + \beta(\boldsymbol{\mu}_{snap} - \boldsymbol{\mu}_{snap}) = \boldsymbol{\mu}_t \tag{37}$$

For each index $i$, denote $\boldsymbol{a}_i := h(\boldsymbol{x}_i, \boldsymbol{\theta}_t)$ and $\boldsymbol{s}_i := h(\boldsymbol{x}_i, \boldsymbol{\theta}_{snap})$ and let

$$\boldsymbol{A} = \frac{1}{n} \sum_{i=1}^{n} \boldsymbol{a}_i = \boldsymbol{\mu}_t$$

and

$$\boldsymbol{S} = \frac{1}{n} \sum_{i=1}^{n} \boldsymbol{s}_i = \boldsymbol{\mu}_{snap}$$

Define the population squared-norm variances by

$$\sigma_a^2 = \frac{1}{n} \sum_{i=1}^{n} \|\boldsymbol{a}_i - \boldsymbol{A}\|^2 \tag{38}$$

and

$$\sigma_s^2 = \frac{1}{n} \sum_{i=1}^{n} \|\boldsymbol{s}_i - \boldsymbol{S}\|^2 \tag{39}$$

and the cross-term

$$\sigma_{as} = \frac{1}{n} \sum_{i=1}^{n} (\boldsymbol{a}_i - \boldsymbol{A}) \cdot (\boldsymbol{s}_i - \boldsymbol{S}) \tag{40}$$

Measuring variance by expected square norm and applying Lemma 6, the variance of the crude estimator is given by

$$Var(\hat{\boldsymbol{\mu}}_t^{crude}) = E\left[\|\hat{\boldsymbol{\mu}}_t^{crude} - \boldsymbol{\mu}_t\|^2\right]$$
$$= E\left[\|\hat{\boldsymbol{\mu}}_t^{crude} - \boldsymbol{A}\|^2\right]$$
$$= \frac{1}{b} \frac{n-b}{n-1} \sigma_a^2$$

The last expression follows from Lemma 6. Similarly, the variance of the control-variate estimator is given by

$$Var(\hat{\boldsymbol{\mu}}_t^{control}) = E\left[\|\hat{\boldsymbol{\mu}}_t^{control} - \boldsymbol{\mu}_t\|^2\right]$$
$$= E\left[\|\hat{\boldsymbol{\mu}}_t^{control} - \boldsymbol{A}\|^2\right]$$
$$= \frac{1}{b} \frac{n-b}{n-1} \left(\sigma_a^2 + \beta^2 \sigma_s^2 - 2\beta \sigma_{as}\right)$$

The variance of the control-variate estimator is a quadratic expression in $\beta$ with the leading coefficient non-negative. Setting the derivative to zero, the minimzer is obtained at

$$\beta^* = \frac{\sigma_{as}}{\sigma_s^2} \tag{41}$$

The variance of the control-variate estimator for optimal choice i.e. for $\beta = \beta^*$ is given by

$$Var(\hat{\boldsymbol{\mu}}_t^{control})|_{\beta=\beta^*} = \frac{1}{b} \frac{n-b}{n-1} \sigma_a^2 (1 - \rho^2) \tag{42}$$

where

$$\rho = \frac{\sigma_{as}}{\sigma_a \sigma_s} \tag{43}$$

For Quantile loss, we have $\|\boldsymbol{a}_i\| = \|\boldsymbol{s}_i\| = 1$ for all $1 \leq i \leq n$. So we have

$$\sigma_a^2 = \frac{1}{n} \sum_{i=1}^{n} \|\boldsymbol{a}_i\|^2 - \|\boldsymbol{A}\|^2 = 1 - \|\boldsymbol{A}\|^2 \tag{44}$$

Similarly

$$\sigma_s^2 = 1 - \|\boldsymbol{S}\|^2 \tag{45}$$

Let

$$c = \frac{1}{n} \sum_{i=1}^{n} \boldsymbol{a}_i \cdot \boldsymbol{s}_i \tag{46}$$

then

$$\sigma_{as} = c - \boldsymbol{A} \cdot \boldsymbol{S} \tag{47}$$

$$\beta^* = \frac{c - \boldsymbol{A} \cdot \boldsymbol{S}}{1 - \|\boldsymbol{S}\|^2} \tag{48}$$

Intuitively, with a small learning rate and frequent updates i.e. if $t - t_{snap}$ is small then $\boldsymbol{a}_i$ and $\boldsymbol{s}_i$ are positively correlated unit vectors i.e. $\boldsymbol{a}_i \cdot \boldsymbol{s}_i \approx 1$. Hence, we have $c \approx 1$ and $\boldsymbol{A} \cdot \boldsymbol{S} \approx \|\boldsymbol{S}\|^2$ i.e. $\beta^* \approx 1$. Hence, the memory-bank estimator is optimal among the control-variate based estimators. The optimal variance is close to zero making the loss-gradient approximation valid even with small batches.

$\square$

### A.3 Quantile Index: Geometric Intuition, Six Blobs and Two-Moons Examples

**Intuition of a Geometric Quantile** In one-dimension each quantile has an associated index between 0 and 1 which can be mean-centered to $[-1, 1]$. The high-dimensional equivalent of the index is a vector in the unit ball in $d$ dimensions. For a probability distribution in $d$ dimensions, every vector in $d$ dimensions is a geometric quantile unlike in one-dimension where the support determines the set of quantiles. Informally, for a given probability distribution, the quantile index of a quantile is given by computing a weighted average of the unit vectors starting at every support point and ending at the quantile, with the weights proportional to the probability. See Fig 5 for a visualization on a toy dataset with 3 points in 2 dimensions. Fig 5a shows the recentered data with the given quantile as the centre. Fig 5b visualizes the unit vectors starting from the data points directed towards the centre in blue and the average of these unit vectors highlighted in red. In other words, a quantile index captures an effective direction of the data from the distribution weighted by the probability mass. The idea of quantile matching is to jointly match all the quantile indices of a specific set of informative quantiles. Geometrically, it is beneficial to choose the support of the dataset as quantiles. The reader may refer to Fig 8 for intuition and the text in subsection 3.2 for practical consideration on informative quantiles.

**A two-dimensional toy example illustrating quantile loss** Consider the following toy example in two dimensions with six classes (see Fig 6a). The training data is given by six blobs each of which is a Gaussian with slightly different number of points, variance-covariance matrices and means. Each blob represents a class. The test dataset is obtained from a linear transformation on the training data. In this example, we set up a bijective mapping between the points in the training and test data to track that the proposed quantile-based approach learns the inverse mapping of the shift. Fig 6d shows the the network approximately learns the inverse linear transform at epoch 1000. For real-data it is not possible to have a bijective correspondence. Hence, we also plot Wasserstein's distance between the test data corrected for covariate shift and the training data. This serves as a distribution-level metric to verify if the quantile-based approach is a proxy for matching probability distributions. See Fig 7 for these plots.

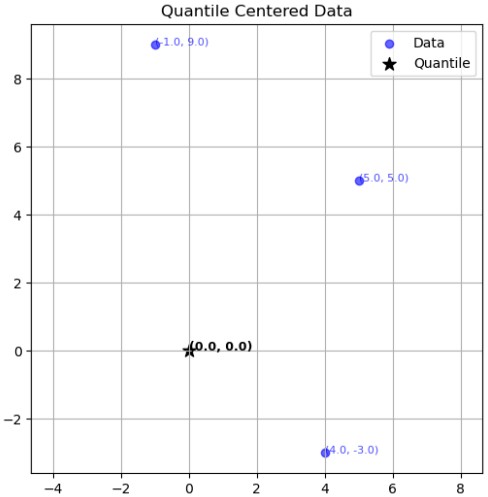
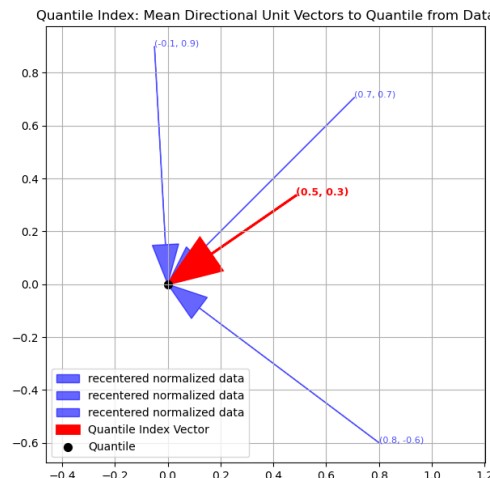

(a) Recentered data points w.r.t the quantile. Data is shown in blue and the quantile is shown in black.

(b) Recentered unit vectors from data to quantile. Quantile index is the average of these unit vectors.

Figure 5: Geometric Visualization of a Quantile Index of a Quantile.

**Another two-dimensional example illustrating a bad initialization** Consider the following toy example in two dimensions with two classes (see Fig 9a). The training data is given by two moons data as shown in Fig 3a. In this example, we again set up a bijective mapping between the points in the training and test data to track the mean-squared error loss. The initialization is closer to a symmetry (rotational invariance of the marginal) as shown in Fig 9a. Figs 9b, 9c and 9d shows the the network approximately learns a transform at epoch 30 such that the marginals match but the class-conditionals flip perfectly. Fig 10 shows that Wasserstein's distance between the decorrupted test data and the training data converges to zero which is consistent with the minimization of quantile loss. This serves as an example to illustrate the importance of a "good" initialization in Theorem 4.

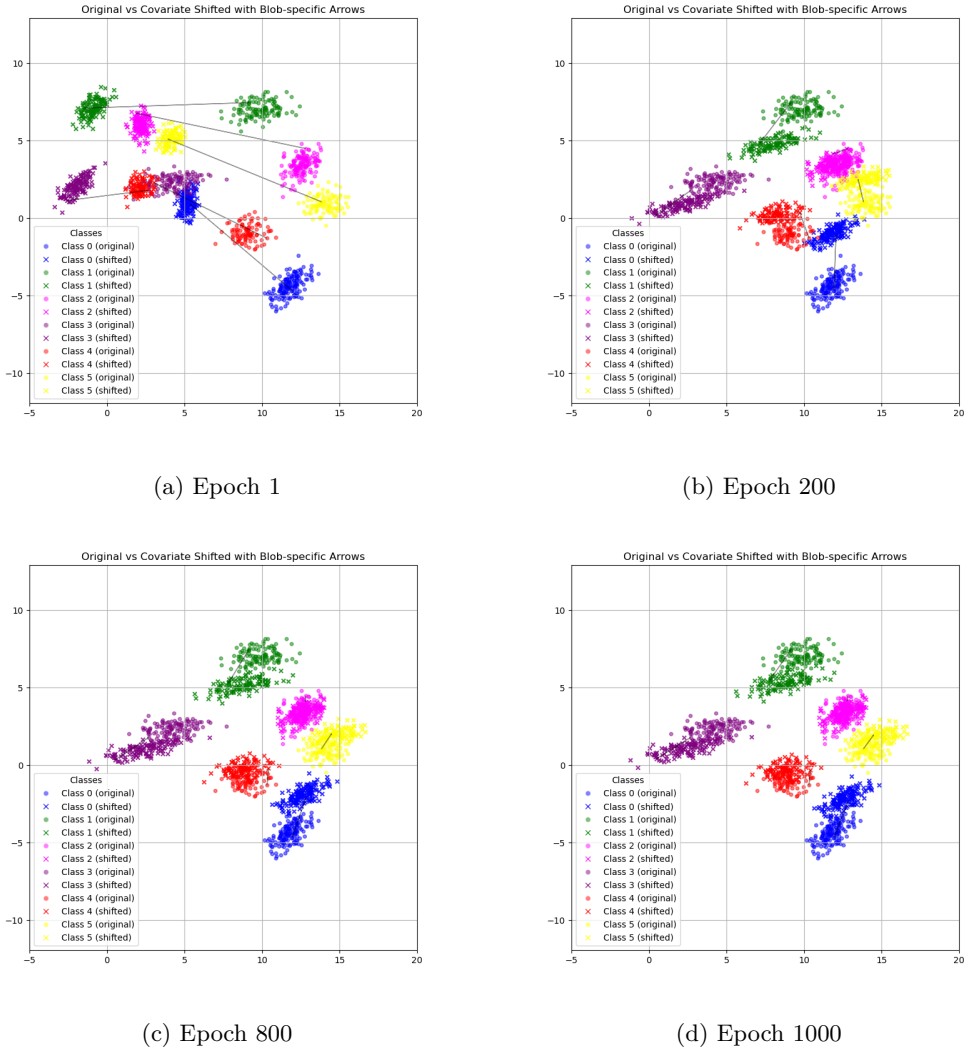

(a) Epoch 1

(b) Epoch 200

(c) Epoch 800

(d) Epoch 1000

Figure 6: (a) - (d) The training and de-corrupted test datasets are overlapped from Epochs 1 to Epoch 1000. The training data corresponds to the points shown in dots. The test data is a covariate-shifted data obtained by a linear transform and is shown by crosses. The colors denote the classes. The arrows denote the cluster-level shifts. As the training progresses, the network learns the shift.

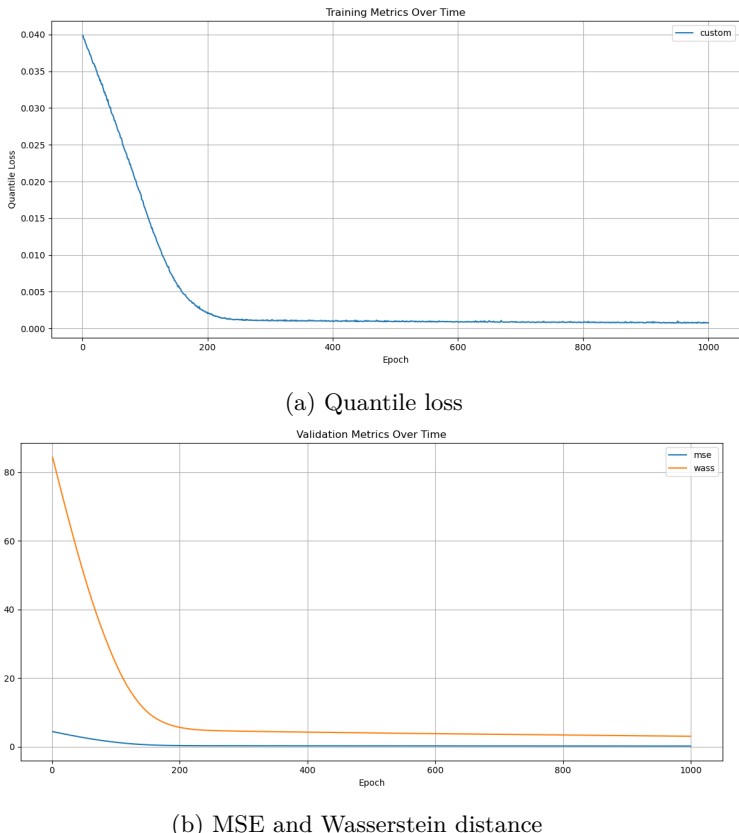

(a) Quantile loss

(b) MSE and Wasserstein distance

Figure 7: (a) Quantile loss between the inverse-mapped test-time covariates and the training data as a function of epochs. The training is done using SGD on a linear transform initialized with identity using quantile loss. (b) Mean squared error between the paired points and the Wasserstein distance between the distributions as a function of epochs. This plot is meant for validation only. Minimizing quantile loss also minimizes the mean-squared error and the Wasserstein's distance between the two probability distributions.

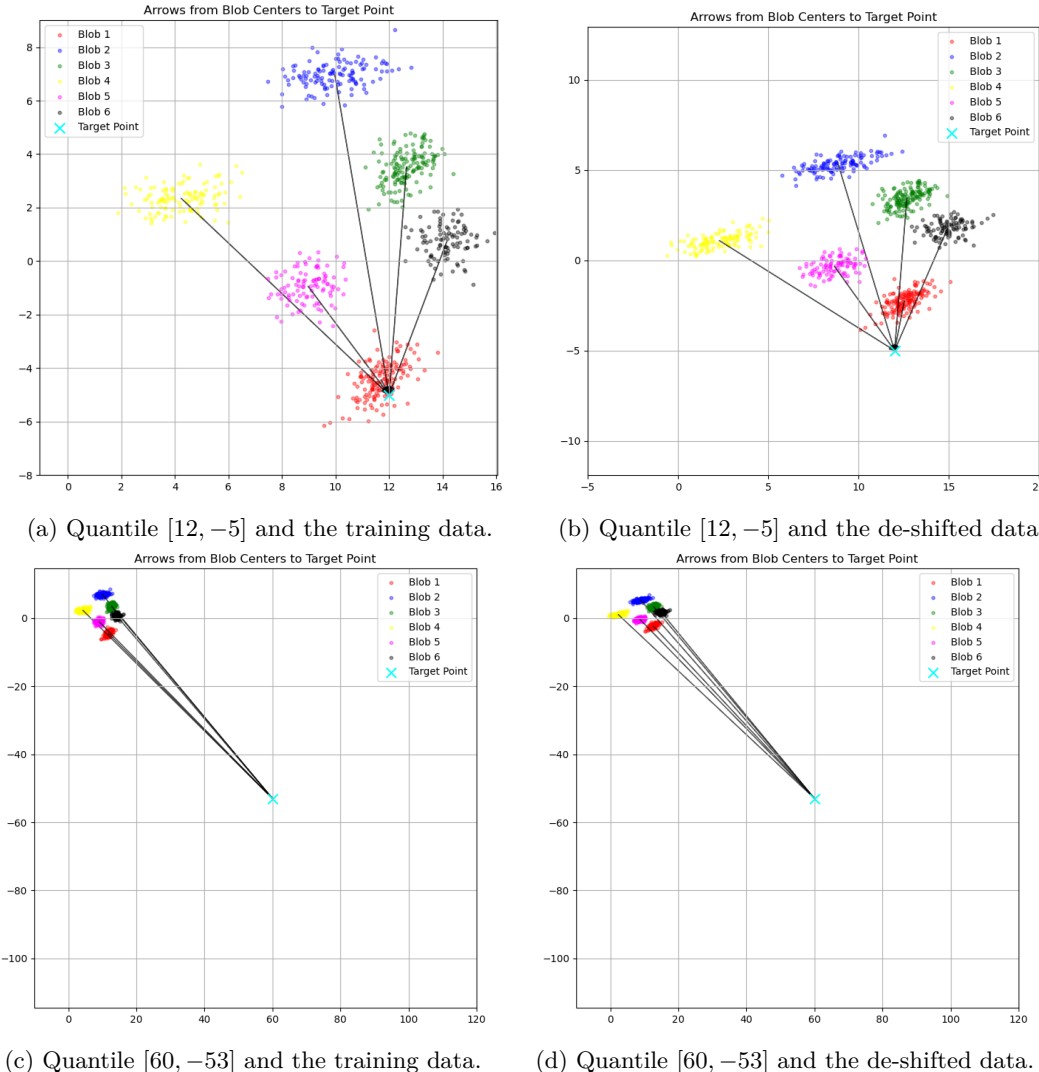

(a) Quantile $[12, -5]$ and the training data.

(b) Quantile $[12, -5]$ and the de-shifted data.

(c) Quantile $[60, -53]$ and the training data.

(d) Quantile $[60, -53]$ and the de-shifted data.

Figure 8: The training dataset corresponds to the points shown in dots in (a) and (c). The de-corrupted test data at the end of 800 epochs is shown by crosses in (b) and (d). The colors denote the classes. The quantile index of a point is approximately the weighted sum of unit vectors originating from the blob centres and ending at the quantile. The weights are the relative number of points in the blob. The quantiles at which the quantile indices are to be matched should be close to the data. The quantile indices of quantiles farther away from data are less sensitive to the overlap of the de-corrupted and the training distribution. The relative norm error of the quantile indices for $[12, -5]$ is 0.64 and for $[60, -53]$ is 0.06 w.r.t. the two datasets

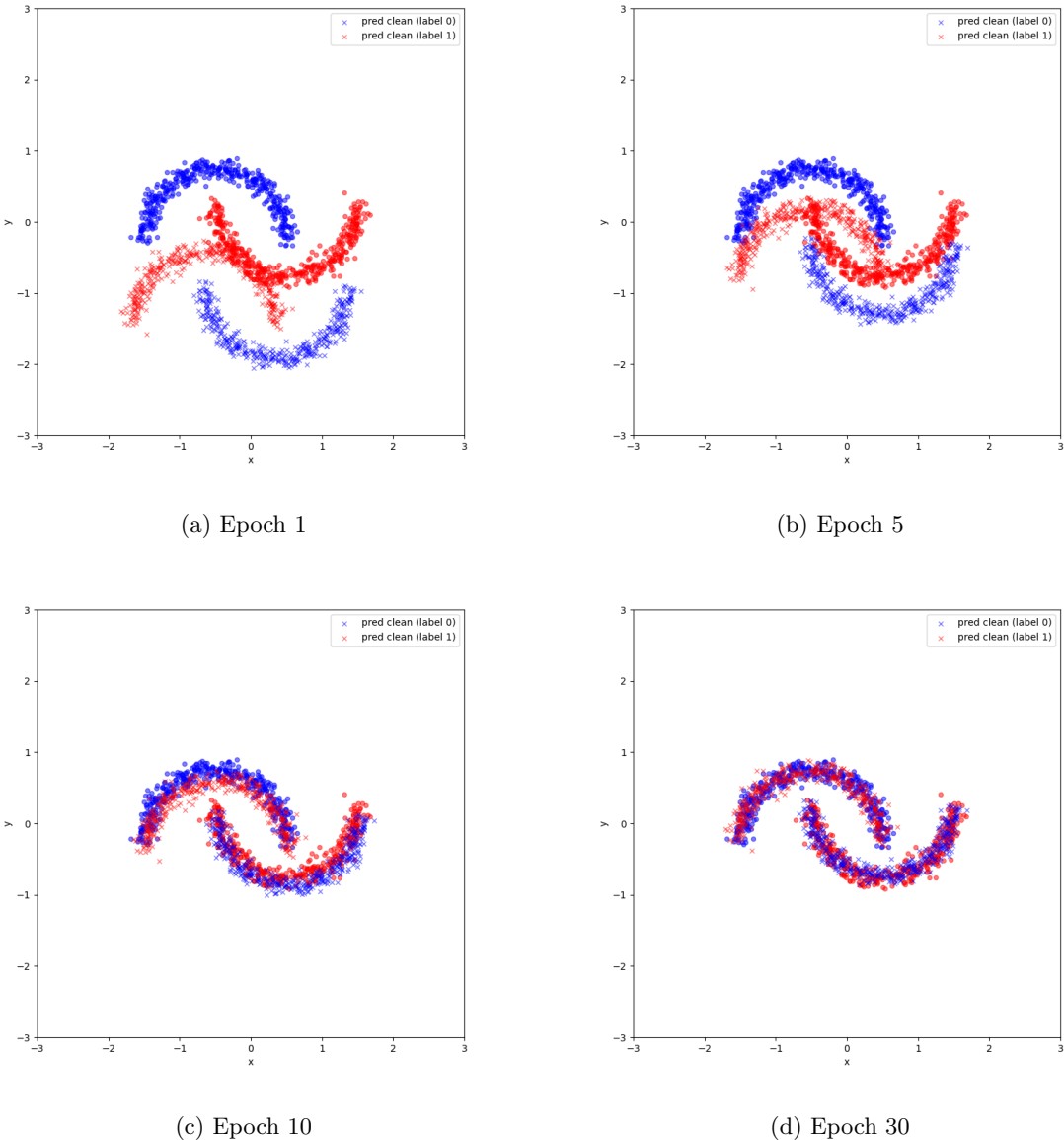

(a) Epoch 1

(b) Epoch 5

(c) Epoch 10

(d) Epoch 30

Figure 9: (a) - (d) The training and de-corrupted test datasets are overlapped from Epochs 1 to Epoch 30. The training data corresponds to the points shown in dots. The test data is a covariate-shifted data obtained by a 180 degree rotation as shown in Fig 3b. The de-corrupted test data is shown by crosses. The The colors denote the classes. As the training progresses, the network matches the marginals better but flips the class-conditionals due to an initialization closer to a minima of quantile loss corresponding to perfectly flipping the class-conditionals.

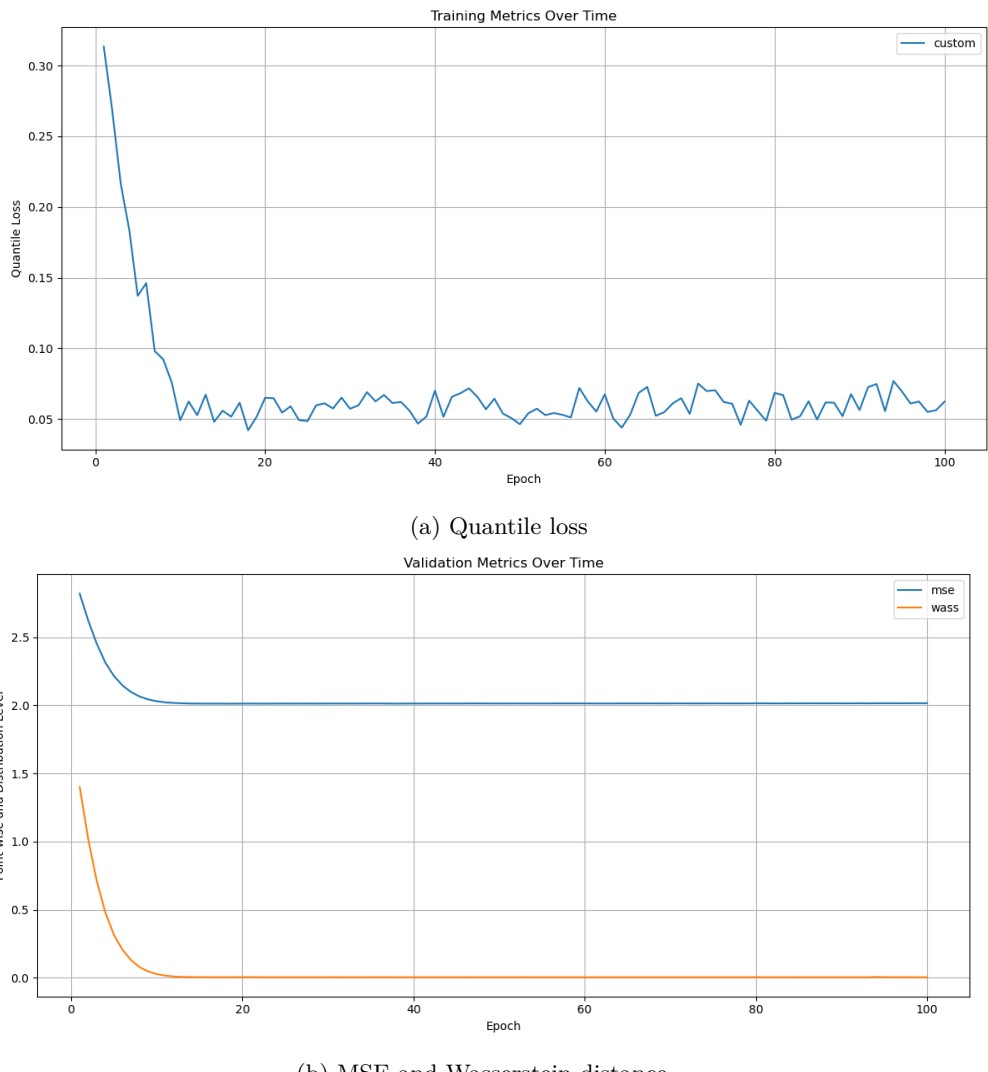

(a) Quantile loss

(b) MSE and Wasserstein distance

Figure 10: (a) Quantile loss between the inverse-mapped test-time covariates and the training data as a function of epochs. The training is done using SGD on a linear transform initialized randomly using quantile loss. (b) Mean squared error between the paired points and the Wasserstein distance between the distributions as a function of epochs. This plot is meant for validation only. Minimizing quantile loss also minimizes the Wasserstein's distance between the two probability distributions but the mean-squared error stagnates as the class-conditionals are swapped.

