# OpenReview forum: "Matching High-Dimensional Geometric Quantiles for Test-Time Adaptation of Transformers and Convolutional Networks Alike"
_TMLR — Rejected by TMLR_

### Review · Reviewer_JGu4 · 2025-11-16

**Summary Of Contributions:**

This paper proposes a test-time adaptation (TTA) method based on geometric quantile loss. It is motivated by the idea that matching geometric quantile indices between decorrupted test features and clean source features can restore clean representations for both CNNs and Vision Transformers. The paper provides a new theorem that claims an equivalence between quantile loss and supervised MSE/optimal transport under certain assumptions. It also introduces a “composite-separable” approximation to make the loss trainable with mini-batches, and evaluates the method on several popular TTA datasets using both CNNs and ViTs.

Strengths:
+ Connecting geometric quantiles and representation alignment is a novel idea to me.
+ The proposed framework includes a memory-bank implementation and some ablations.
+ Empirical results show improvements over baseline models on standard corruption benchmarks.

Weaknesses:
- Several assumptions in the paper are not correct and/or practical. More below.
- Figure 4 does not provide empirical evidence for Theorem 4. It merely implies correlation between quantile loss and MSE.
- Although Section 5 shows improvement, there's no evidence that it come from Theorem 4.
- Composite separability and the memory-bank approximation are not well justified. More below.
- The datasets are too small and simple. They should be used to validate Theorem 4, not to evaluate the practicality of the proposed method. We need bigger sets like CoTTA.
- There are no numbers on computational cost, despite the method’s high complexity.
- The paper includes a lot of math without enough context, making the theoretical parts harder to follow given the simpler underlying idea. More below.

**Additional Comments:**

The proposed method assumes access to source features at test time. This is not a standard assumption in many TTA settings.

**Audience:**

Yes

**Audience Explanation:**

TTA is an active area, and the idea of using geometric quantiles for feature alignment may interest some people in the community.

**Broader Impact Concerns:**

No.

**Claims And Evidence:**

No

**Claims Explanation:**

Neither the empirical nor the theoretical evidence convincingly supports the main claims of the paper

• Theorem 4 is the main contribution, but it relies on several strong assumptions that don’t hold in typical TTA settings.

    – For example, the paper says “Covariate-shift allows us to assume a joint distribution on (X^+, X) such that T^*(X^+) = X with probability 1.”
    This basically assumes the existence of a perfect decorruption operator. Covariate shift does not imply that — it only says the label distribution stays the same. I’m willing to pretend to accept it for the sake of review, but I don’t think it’s justified.

    – A3a (identifiability) is another example. It assumes that if two models produce the same output distribution, then they produce the same output value for almost every input. This is very strong. It’s easy to build counterexamples on symmetric domains (like two moons or any symmetric distribution) where different functions give the same distribution but different pointwise outputs.

• Figure 4 does not validate Theorem 4.
It just shows that quantile loss and MSE are correlated on CIFAR-10, which is expected for many distance-based metrics. It doesn’t show equivalence. There are no controlled experiments where the assumptions hold. For example, the corruptions in CIFAR10-C/ImageNet-C are not invertible, unlike the assumption in the paper that they are.

• Section 5 shows some improvements, but it doesn’t show that those improvements come from the theory or from the quantile-loss objective.
It only shows the method can help in practice, which is different from confirming the theoretical claims.

• Composite separability and the memory-bank approximation are not really justified.
There’s no error analysis, no convergence analysis, and no ablations testing whether they matter.

**Requested Changes:**

My main concerns are Theorem 4 and experiments.

* Revisit the core assumptions behind Theorem 4, which do not hold in typical TTA settings. Either justify them or relax them. Otherwise, I’m willing to accept them for the sake of review, but they need to be addressed.

* Provide controlled experiments where the assumptions of Theorem 4 actually hold. For example, use simple synthetic invertible transforms like a rotation to test whether the theoretical claims are correct.

* Clarify the role of Theorem 4 in the method. Explain whether the practical gains really depend on the theoretical result, or if the theory is mostly independent of the actual implementation.

* Justify composite separability and the memory-bank approximation. Add some explanation, analysis, or experiments showing that these choices are necessary or at least reasonable.

* Run the pipeline on larger, more practical TTA benchmarks.

* Add computational cost (runtime, memory), since the method is not lightweight.

I have many other comments and suggestions, but I’d like the authors to address these critical points first.

---

> ### Author Response · Authors · 2025-11-29
>
> We sincerely thank the reviewer for their thorough and constructive feedback. We appreciate the recognition that our core idea of connecting geometric quantiles to representation alignment is novel and that our empirical results show meaningful improvements.
>
>
> 1) *Several assumptions in the paper are not correct and/or practical. Revisit the core assumptions behind Theorem 4, which do not hold in typical TTA settings. Either justify them or relax them. Otherwise, I’m willing to accept them for the sake of review, but they need to be addressed.*
>
> We partially agree with the reviewer in that the writing does not convey the intention well. We thank the reviewer for pointing this out.
>
> We do not assume perfect reconstruction, rather an approximate one. We make the following edits in section 2.1 to reflect this --
>
>
> there exists $T^{\*}$   such that  $X = T^{\*}(X^{+}) + \epsilon$ such that $E[\epsilon | X] = 0$ and $Cov(\epsilon | X) = \Sigma$ such that the operator norm $\| \Sigma \|_{op} \leq \delta$ (maximum of the modulus of Eigenvalues) for some small $\delta > 0$.  As a result, the squared norm error $E \left[ \| X - T^{*}(X^{+}) \|^2 \right] \leq d \delta$ where $d$ is the dimension of $X$ and $X^{+}$.
>
> Also, theorem 4 has two parts.
> - The first part (statements (a),(b),(c) ) assumes an idealistic scenario where a perfect decorruptor exists.
> - The second part (statements (d),(e),(f)) of the theorem 4 assumes a more realistic scenario where the mean-squared error can be reduced to an error of $\Delta_{1}(\eta)$.
>
> Similarly, the identifiability assumption (A3a) is relaxed to hold within a small ball around the minimizer and not globally. (A3b). We modified the reconstruction assumption (A2a) in part B to reflect the changes made in the assumptions in Section 2.1 (see A2b).
>
> Since, pre-trained classifiers applied to covariate-shifted data achieve non-trivial classification accuracies (much greater than random classifier), identity initialization acts as a point within this small ball. Since the pre-trained classifiers are usually well-trained, the clusters across classes are well-separated making the local-identifiability assumption realistic. The equivalence statements in Part B are modified to hold for $\eta > \sigma^2$ i.e. for any value above the irreducible error.
>
> 2) *Figure 4 does not provide empirical evidence for Theorem 4. It merely implies correlation between quantile loss and MSE. Provide controlled experiments where the assumptions of Theorem 4 actually hold. For example, use simple synthetic invertible transforms like a rotation to test whether the theoretical claims are correct.*
>
> Note that an empirical proof of core part of Thm 4 constitutes the following -- (i) If we minimize quantile loss then MSE should be minimized and (ii) conversely if we minimize the MSE then quantile loss should be minimized.
>
> Fig 4b shows that if we minimize the quantile loss then MSE is minimized -- showing (i). We have not explicitly verified the reverse implication (ii), since it is easy to see from Thm 4 that minimizing MSE makes the distributions closer and consequently quantile loss is minimized.
>
> Also, Fig 6 in the Appendix verifies the same in a controlled setting: the original data is 6 blobs in 2 dimensions and the covariate shifted data is a rotation of the original data. It is visually clear from Fig 6d that minimising the quantile loss minimizes MSE. Additionally, Fig 7 shows the raw mean-squared error and Wasserstein's distance as a function of the epochs confirming the same.
>
> A possible source of confusion is from the explanation of Fig 4, where it was not explicitly stated that -- we indeed minimized quantile loss and observed that MSE loss is reducing as well. We edited the text explaining Fig 4 in the revised version to reflect this.
>
> 3) *Although Section 5 shows improvement, there's no evidence that it come from Theorem 4. Clarify the role of Theorem 4 in the method. Explain whether the practical gains really depend on the theoretical result, or if the theory is mostly independent of the actual implementation.*
>
> To restate -- The main contribution of the article is an **architecture agnostic approach for correcting distribution shifts**, and we propose a framework based on quantiles to achieve this
>
> - Theorem 4 shows the theoretical justification for the proposed framework -- We are able to minimize pairwise losses **without** access to the paired samples. This is the implication of Theorem 4 part B.
> - Experiments in Section 5 verify this empirically by applying our method across architectures.
>
> Hope the above explanation clarifies the role of theorem 4 and experiments in section 5 within the scope of our quantile loss framework.

---

> > ### Author Response · Authors · 2025-11-29
> >
> > 4) *Composite separability and the memory-bank approximation are not well justified. Justify composite separability and the memory-bank approximation. Add some explanation, analysis, or experiments showing that these choices are necessary or at least reasonable.*
> >
> > We partially agree with the reviewer. However, note that the composite-separability is a property of the quantile-loss as defined in our article.
> > - For a separable loss function, implementing mini-batch SGD is well-known [1].
> > - On the other hand, for generic non-separable loss functions, implementing mini-batch SGD is non-trivial and computationally expensive.
> >
> > For composite separable loss functions, inspired by the idea of control variates (ref Ross Simulation textbook 2022) for variance reduction (similar to SAGA and SVRG - see [2]), we introduce the notion of memory bank approximation. To elaborate, Eq 19 is a control-variate based variance reduced estimator of the quantile index. So, in essence with a small batch we are effectively approximating the gradient of the full loss by using a combination of current batch and recent non-batch samples. The choice of the control-variate is very similar to SVRG and the proof can be made rigorous in similar lines.
> >
> > For brevity, we did not consider providing a proof earlier. In the revised version, we added a sketch of proof in the appendix A.2. Also, we added more explanation in the main article.
> >
> > [1] Stochastic Gradient Descent on Separable Data: Exact Convergence with a Fixed Learning Rate, Mor Shpigel Nacson, Nathan Srebro, Daniel Soudry, AIStats 2019
> > [2] Accelerating stochastic gradient descent using predictive variance reduction, Rie Johnson, Tong Zhang, NeurIPS 2013)
> >
> >
> > 5) *The datasets are too small and simple. They should be used to validate Theorem 4, not to evaluate the practicality of the proposed method. We need bigger sets like CoTTA. Run the pipeline on larger, more practical TTA benchmarks.*
> >
> > If our understanding is correct, the reviewer wanted to check practicality on larger datasets.
> >
> > In this regard, we wish to reiterate - the key contribution of this article is to illustrate that one can learn the decorruptors even without access to pairwise samples. This is the scope of the experimental section (Section 5) and the main contribution which is Thm 4.
> >
> > 6) *There are no numbers on computational cost, despite the method’s high complexity. Add computational cost (runtime, memory), since the method is not lightweight.*
> >
> > **Memory Overhead**: Our method introduces two components beyond the frozen classifier: (i) a quantile loss module with memory bank, and (ii) a decorruptor network.
> >
> > For a representative setup (CIFAR100C with ResNet18, 10,000 source features of dimension 512, batch size 128),
> > - the quantile-loss module requires approximately 570 MB on an Nvidia H100.
> > - The 6.2M parameter decorruptor increases peak memory from 765 MB (classifier only) to 1429 MB (total), representing an overhead of 664 MB.
> >
> > This is comparable to mean-teacher-based TTA methods, which duplicate the entire classifier. Importantly, *our memory overhead  (due to decorruptor) scales only depends on the complexity of the decorruption* and *not on the complexity of classification* -- This implies a fixed memory overhead *irrespective of the complexity of the underlying classifier*
> >
> > **Time Overhead:**
> > - The quantile-loss computation adds approximately 2 ms per batch.
> > - End-to-end training time increases from 1.44s to 2.03s per epoch with the decorruptor,
> >
> > This too is competitive with existing TTA methods while providing the advantage of not requiring paired clean-corrupt data for training.
> >
> > We have added a dedicated computational cost analysis to the revised manuscript. We hope this clarifies that our method maintains practical efficiency despite its algorithmic sophistication.
> >
> >
> >
> > 7) *The proposed method assumes access to source features at test time. This is not a standard assumption in many TTA settings.*
> >
> > While this assumption is not a standard assumption, it is not a strong assumption either. We added the following  remarks in Section 2.1 to reflect the same.
> >
> > ```
> > The requirement on access to the features generated by the classifier on the training data is a mild assumption. Privacy concerns on the training data does not arise as our method requires only the probability distribution of these features. While the existing approaches do not use these features in their methods, we claim that this information is useful at the cost of a small memory overhead for storing the source features.
> > ```

---

> > > ### Comment · Reviewer_JGu4 · 2025-12-01
> > > **5. The datasets are too small and simple.**
> > >
> > > Is there any plans to include additional results from larger datasets during the review period?

---

> > > > ### Author Response · Authors · 2025-12-01
> > > >
> > > > Unfortunately, given the constraint on computational resources it might not be possible to test the method on large datasets such as imagenetc during the review period.
> > > >
> > > > Regarding the complexity of datasets, as tables 1 and 2 show, the value of our contribution is reflected in the architecture adaptability to transformers and CNNs alike, a trait which current SOTA is unable to exhibit. The experiments and results align with the scope described earlier, which are consistent with our central hypothesis.
> > > >
> > > > Since there were no further comments on the earlier concerns, we hope we have addressed all of them satisfactorily. Kindly let us know otherwise.

---

### Review · Reviewer_QHLg · 2025-11-20

**Summary Of Contributions:**

The main contribution of this paper lies in the use of the quantile loss function to measure the discrepancy between given initial data and its corrected version after corruption. In this regard, Theorem 4 states that the expected pairwise mean-square error and the quantile loss function vanish simultaneously when the initial data and the corrected data coincide. In other words, the authors advocate using the quantile loss function to assess similarity rather than relying on the mean-square loss function.

Although the problem addressed is interesting from both mathematical and practical perspectives, the referee finds the mathematical development rather weak. The referee apologizes for being blunt, but Lemma 5 and Proposition 1 rely on elementary linear-algebra, and Theorem 4 follows immediately from Proposition 1. Moreover, Proposition 1 is proved only for finite discrete measures, while the authors merely assert—without providing a proof—that the general case is analogous, despite requiring more rigor. Of course, this does not diminish the potential importance of the results for the machine-learning community.

**Additional Comments:**

No additional comment.

**Audience:**

Yes

**Audience Explanation:**

The problem dealt with in this paper is quite important for image classification since the training data experience corruption and needs to be cleaned as much as possible before test-time. At the mathematical level, the problem is also interesting yet seems difficult. Indeed, one needs to find a map pushing the corrupted data onto the original one.

**Broader Impact Concerns:**

There is no concern.

**Claims And Evidence:**

Yes

**Claims Explanation:**

The paper encloses a considerably large amount of numerical evidences.

**Requested Changes:**

The paper is well written and the motivation is well explained.

---

> ### Author Response · Authors · 2025-11-29
>
> We sincerely thank the reviewer for the kind comments and recognizing the potential importance of the main result.
>
> *... Lemma 5 and Proposition 1 rely on elementary linear-algebra, and Theorem 4 follows immediately from Proposition 1 .. Proposition 1 is proved only for finite discrete measures..."
>
> We partially agree with the reviewer in that -- the mathematical statements are not the most  general statements possible, but are specialized to the case we are interested in this article.
>
> However, the aim of the article was to provide a mathematical framework *to learn pairwise distances without access to the pairwise samples*. The theorem and experiments were meant as justification and to provide intuition for the proposed approach.
>
> We consider a detailed mathematically complete statement as future work. We modified the statement after Proposition 1 to reflect that the proof is provided only for the finite support case.

---

### Review · Reviewer_QKLP · 2025-11-21

**Summary Of Contributions:**

This paper proposes a novel architecture-agnostic test-time adaptation (TTA) method to address the performance degradation of models when faced with covariate shifts (such as image corruption). Most existing TTA methods rely on specific model architecture components (especially batch-norm layers such as TENT and SoTTA), making them difficult to apply directly to modern architectures using layer-norm or group-norm structures like VITs. Instead of modifying the pre-trained and frozen classifier, our solution adds a "de-corruption" adapter network ($T_{\theta}$) before the classifier. This adapter is trained at test time and is designed to map corrupted target images (test data) back to the distribution of the source data (training data). The core innovation of this paper lies in proposing a novel "quantile loss" to train this adapter. Since there are no paired (damaged-clean) samples during testing, this method instead operates in a high-dimensional feature space (using the feature extractor $f$ of the frozen classifier). It achieves distribution alignment by matching the high-dimensional geometric quantiles of the source data features $f(X)$ and the "de-damaged" target data features $f(T_{\theta}(X^{+}))$.

Key Strengths:
Different from previous BN-dependent methods, this method does not rely on BN layers. Experiments (Table 1) demonstrate that it is applicable to both classic CNNs (ResNet18) and various Transformer architectures, achieving performance improvements in all cases.

The paper provides theoretical support for quantile loss (Theorem 4). This theorem shows that, under conditions of "good" initialization (such as identity mapping), minimizing the quantile loss (matching marginal distributions) is equivalent to minimizing the mean squared error (MSE) loss requiring paired samples. This provides strong theoretical support for aligning class-conditional distributions even without labels.

Introducing high-dimensional geometric quantiles as a distribution matching tool into the TTA field is a novel and insightful idea.The paper analyzes the "composite-separability" of the loss function and proposes an efficient mini-batch training algorithm using a memory bank, solving the training challenge of non-sample-separable loss.

Key Weaknesses:
Compared to current state-of-the-art SOTA methods like SoTTA, this approach still lags behind ResNet18 (e.g., on CIFAR100C, SoTTA achieves 60.5%, while our method achieves 49.9%). Although the authors outperform SoTTA-ResNet18 with CCT (65.0%), this is not a direct comparison of similar architectures.

This method assumes access to unlabeled source data (or its features) during the TTA stage. While the authors argue this is reasonable in many applications, it is a higher requirement than for "fully test-time" methods like TENT or SoTTA.

The adapter itself is a CNN with 6.2 million parameters. This introduces additional computational overhead during inference. The paper does not explicitly compare its overhead with methods such as SoTTA(which only fine-tunes the statistics of the BN layer).

Experiments are conducted on small-scale datasets, while larger datasets such as ImageNet-C are not considered for validation of the proposed approach.

**Audience:**

Yes

**Audience Explanation:**

Domain adaptation is a very active and practically crucial research area. A recognized limitation is that most SOTA TTA methods cannot be applied to ViT architectures. This paper proposes a novel, theoretically supported, architecture-agnostic solution that directly addresses this gap. Researchers and practitioners interested in TTA, Domain adaptation, model robustness, and ViT architectures will find this work highly valuable.

**Broader Impact Concerns:**

This paper focuses primarily on improving the robustness of machine learning models under distributional shifts. This is a technical contribution aimed at making AI systems more reliable in the real world, such as in the face of weather changes or sensor malfunctions. I did not foresee any significant negative ethical or social impacts from this research.

**Claims And Evidence:**

Yes

**Claims Explanation:**

The core argument of this paper is that its TTA method is "architecture-independent." This is supported by ample experimental evidence. Table 1 clearly shows that the method achieves significant accuracy improvements compared to the baseline on ResNet18 and three different Transformer architectures: CCT, CVT, and ViT.

Furthermore, the paper's theoretical argument (Theorem 4: equivalence of quantile loss and pairwise MSE loss) is empirically validated in Figure 4. Figures 4a and 4b show a strong positive correlation between quantile loss and MSE loss (calculated using pairwise data) during training, and both decrease with increasing training steps.

The experimental settings (using CIFAR10-C, CIFAR100-C, and TinyImageNet-C datasets) are standard benchmarks in the TTA field, while larger datasets such as ImageNet-C are not experimented. .

**Requested Changes:**

Currently, in Table 1, the results on ResNet18 (e.g., 79.4% on CIFAR10C) are lower than SoTTA (82.2%).The authors should more explicitly acknowledge this in the discussion section and emphasize the value of their method in its "architecture independence," a flexibility sufficient to compensate for the slight performance loss on CNNs. Simultaneously, they should more prominently highlight that on CIFAR100C, CCT-Quantile (65.0%) indeed outperforms SoTTA-ResNet18 (60.5%), strongly demonstrating the potential of the new architecture + new method.

Experiments on larger benchmarks such as ImageNet-C will strengthen the experimental validation of the work.

This method requires access to the source data (or its features) to compute the quantile index ($U_{\mu^{f}}$). The TTA setting of this paper should be more clearly defined in the introduction or related work section. It is not entirely "source-free". Please add a paragraph comparing it to TTA methods that do not require source data at all (such as TENT, EATA), discussing the practical impact and trade-offs of this dependency.

Adding a 6.2M parameter adapter incurs additional inference and (one-time) TTA training overhead. Add a brief analysis (or a table in the appendix) discussing: (a) the TTA time required to train the adapter (e.g., on CIFAR10-C); and (b) how it compares to the overhead of updating parameters (such as BN statistics or affine parameters) at test time using methods like SoTTAor TENT.

Theory (Theorem 4) depends on "good" initialization (assumption A3b). An identity map was used as initialization in the experiments. Please add an ablation experiment (or discuss in the appendix) to see how the performance would be if the adapter $T_{\theta}$ were randomly initialized. This would help verify the importance of identity mapping as a “good” initialization and enhance the credibility of the theory in practice.

---

> ### Author Response · Authors · 2025-11-29
>
> We sincerely thank the reviewer for the kind comments, thorough constructive feedback and recognizing the potential of high-dimensional geometric quantiles as a distribution matching tool.
>
>
>
> 1. *Rewrite to better emphasize "architecture independence" and it's flexibility to compensate slightly loss on CNNs*
>
> We included the following text in the revised version:
>
>
> Although our ResNet18-based results fall short of SoTTA on both CIFAR10C and CIFAR100C ($79.4$% vs. $82.2$% and $49.5$% vs. $60.5$%, respectively), the strength of our approach lies in its architecture independence. Unlike SoTTA, which is tailored to CNNs, our method applies seamlessly to transformer architectures which are widely known to surpass CNN-based classifiers in performance. This flexibility is not merely conceptual: when instantiated on CCT, our method achieves $65.0$% on CIFAR100C, outperforming SoTTA’s ResNet18 result of $60.5$%. The increase of $6.6$% accuracy on CIFAR10C and $4.5$% on CIFAR100C compared to the current SOTA is attributed to the applicability of quantile-loss minimization to transformer architectures. Thus, while there is a slight performance gap on CNNs, our method compensates for it through broader applicability and demonstrably stronger performance on more advanced architectures, underscoring its long-term potential.
>
>
>
> 2. *The method is not source-free...* and *Clarify the TTA setting and compare with other TTA methods that do not require source data*
>
> This is indeed a valid argument, and thank the reviewer for pointing it.
>
> Our argument for this is -- Since source data is usually available, when training, *why not use the extra information for our benefit?* Note the following:
> - We only need features at the last layer, which can be obtained even after the training.
> - We only need "distributional-equivalence" -- So, the features need not be exactly be the training data. Hence, privacy concerns are alleviated. In other words, the source data cannot be reconstructed from the features and the classifier. In that sense, we are source-free
> - The memory overhead is also negligible.
>
> Nevertheless, we agree that these should be pointed out explicitly in the article and have made the necessary changes.
>
>
>
> 3. *Time required to train TTA adapter* and *comparison with other methods*
>
> **TTA Training Time for the Adapter:**: For our representative setup (CIFAR100C with ResNet18), training the 6.2M parameter decorruptor adds an overhead of *0.59s per epoch (from 1.44s to 2.03s)*. This is a one-time training cost incurred offline before test-time deployment. Once trained, the decorruptor adds approximately *2 ms per batch* during inference for the quantile-loss computation and forward pass through the adapter network.
>
> **Comparison with other approaches**: Methods like TENT and SoTTA update only batch normalization statistics during test time, incurring negligible computational overhead. In contrast, methods like CoTTA that maintain teacher-student architectures duplicate the entire network, resulting in approximately twice the memory and computational cost. Our method falls between these extremes, with overhead comparable to CoTTA but fundamentally different in capability.
>
> However, we emphasize a critical distinction -- neither TENT, SoTTA, nor CoTTA learn a decorruptor in the absence of paired clean-corrupt samples - this is the core contribution of our work. Learning an explicit decorruptor provides several key advantages beyond computational considerations: (i) Interpretability: the decorruptor provides an explicit, inspectable transformation that removes corruption, rather than implicit adaptation through statistics or ensemble predictions; (ii) Favorable scaling: our overhead depends only on the complexity of the corruption (i.e., decorruptor architecture), not the classifier complexity. For large-scale classifiers (e.g., vision transformers with hundreds of millions of parameters), methods like CoTTA scale linearly with classifier size, while our overhead remains constant as long as the corruption complexity is unchanged.
>
>
> 4. *Experiment demonstrating bad initialization*
>
> We thank the reviewer for pointing this experiment to validate the usefulness of a good initialization. We added an ablation experiment using the two-moons example in appendix A.3 and figure 9. In this experiment, we initialize the adapter closer to the rotation-invariance of the marginals. As a result, quantile loss minimization in turn matches the marginals but perfectly flips the class-conditionals. Note that is also minima of Quantile loss but does not minimize MSE.

---

### Author Response · Authors · 2025-11-29

We sincerely thank all the reviewers for their quick, thorough and constructive feedback. We appreciate the recognition that the idea of using geometric quantiles to TTA is novel and our empirical results show meaningful improvements.

The changes made have been highlighted in red in the revised article.

---

### Decision · Action_Editor_tVLy · 2025-12-18

**Recommendation:** Reject

**Additional Comments:**

While the paper explores an interesting direction, the current submission does not yet meet the required standard in terms of theoretical soundness and empirical validation. The contribution relies heavily on a theoretical result whose assumptions remain strong and only partially justified for realistic TTA settings (based on elaborate reviewer feedback), leaving a gap between the theory and practical applicability. Experimental validation is limited to relatively small benchmarks, with no large-scale evaluation to establish robustness or broader impact, and resource constraints prevented addressing this concern during rebuttal. As a result, despite the novelty of the idea, the work is not yet ready to support acceptance at this stage. I, however, recommend considering a re-submission after a major revision, in which the aforementioned shortcoming are meaningfully addressed.

**Audience:**

Yes

**Audience Explanation:**

Test-time adaptation (TTA) is a relevant research area, and the idea of using geometric quantiles for architecture-agnostic adaptation is interesting to explore. Researchers interested in robustness, distribution shift, and TTA methods would most probably find the proposed approach and its underlying ideas worth knowing, even though the current evidence is not yet sufficient to support acceptance.

**Claims And Evidence:**

No

**Claims Explanation:**

While the paper presents an interesting idea and includes both theoretical analysis and empirical results; however, the evidence does not sufficiently or convincingly support the central claims. As is highlighted by the reviewers, the main theoretical contribution relies on strong assumptions (e.g., approximate decorruptability and identifiability) that are not clearly justified for realistic test-time adaptation settings, and the connection between the theory and practical performance remains weak. The evaluation is limited to small-scale benchmarks, with no large-scale experiments to demonstrate robustness or general applicability, and the results do not meaningfully isolate or validate the role of the proposed theoretical insights.

**Resubmission Of Major Revision:**

The authors may consider submitting a major revision at a later time.